# Developing a clinically relevant radiosensitizer for temozolomide-resistant gliomas

Radu O. Minea[1], Tuan Cao Duc[2], Stephen D. Swenson[1], Hee-Yeon Cho[1], Mickey Huang[3], Hannah Hartman[4], Florence M. Hofman[5], Axel H. Schönthal[4], Thomas C. Chen[1,5]*

**1** Department of Neurological Surgery, Keck School of Medicine (KSOM), University of Southern California (USC), Los Angeles, California (CA), United States of America, **2** Haiphong University School of Pharmacy, Haiphong, Vietnam, **3** Broad Center for Regenerative Medicine and Stem Cell Research, University of Southern California, Los Angeles, CA, United States of America, **4** Department of Molecular Microbiology and Immunology, Keck School of Medicine, University of Southern California, Los Angeles, CA, United States of America, **5** Department of Pathology, Keck School of Medicine, University of Southern California, Los Angeles, CA, United States of America

* Thomas.Chen@med.usc.edu

**Data Availability Statement:** All relevant data are within the paper and its Supporting Information files.

**Funding:** This study was funded in part by the Hale Family Research Fund and Sounder Foundation

## Abstract

The prognosis for patients with glioblastoma (GB) remains grim. Concurrent temozolomide (TMZ) radiation—the cornerstone of glioma control—extends the overall median survival of GB patients by only a few months over radiotherapy alone. While these survival gains could be partly attributed to radiosensitization, this benefit is greatly minimized in tumors expressing $O^6$-methylguanine DNA methyltransferase (MGMT), which specifically reverses $O^6$-methylguanine lesions. Theoretically, non-$O^6$-methylguanine lesions (i.e., the N-methylpurine adducts), which represent up to 90% of TMZ-generated DNA adducts, could also contribute to radiosensitization. Unfortunately, at concentrations attainable in clinical practice, the alkylation capacity of TMZ cannot overwhelm the repair of N-methylpurine adducts to efficiently exploit these lesions. The current therapeutic application of TMZ therefore faces two main obstacles: (i) the stochastic presence of MGMT and (ii) a blunted radiosensitization potential at physiologic concentrations. To circumvent these limitations, we are developing a novel molecule called NEO212—a derivatization of TMZ generated by coupling TMZ to perillyl alcohol. Based on gas chromatography/mass spectrometry and high-performance liquid chromatography analyses, we determined that NEO212 had greater tumor cell uptake than TMZ. In mouse models, NEO212 was more efficient than TMZ at crossing the blood-brain barrier, preferentially accumulating in tumoral over normal brain tissue. Moreover, in vitro analyses with GB cell lines, including TMZ-resistant isogenic variants, revealed more potent cytotoxic and radiosensitizing activities for NEO212 at physiologic concentrations. Mechanistically, these advantages of NEO212 over TMZ could be attributed to its enhanced tumor uptake presumably leading to more extensive DNA alkylation at equivalent dosages which, ultimately, allows for N-methylpurine lesions to be better exploited for radiosensitization. This effect cannot be achieved with TMZ at clinically relevant concentrations and is independent of MGMT. Our findings establish NEO212 as a superior radiosensitizer and a potentially better alternative to TMZ for newly diagnosed GB patients, irrespective of their MGMT status.

(provided to T.C.C), and by generous support from NeOnc Technologies, Inc. (Los Angeles, CA). The funders had no role in study design, data collection and analysis, decision to publish, or preparation of the manuscript.

**Competing interests:** I have read the journal's policy and the authors of this manuscript have the following competing interests: T.C.C. is the founder of NeOnc Technologies, Inc., a startup company formed to develop derivative drug-conjugate solutions for intranasal, oral, and systemic administration in multiple clinical indications. R.O. M., S.D.S., and T.C.C. are also co-founders of Disintegrin Therapeutics, Inc., a startup company aimed to develop disintegrin-based theranostics for multiple clinical indications. This does not alter our adherence to PLOS ONE policies on sharing data and materials.

## Introduction

Glioblastoma (GB), also known as grade IV astrocytoma, is the most malignant type of glioma and a devastating brain cancer affecting both genders and all ages [1]. Despite important advances in surgical techniques, imagistic modalities and computer-assisted stereotactic delivery of radiotherapy, the overall median survival for patients with GB is still about 14 months from the time of diagnosis [2,3]. The gold standard of post-operative therapy for newly diagnosed patients with primary GB was first introduced more than a decade ago and is called the Stupp protocol [4]. This consists of a regimen of conformal radiotherapy (fractionated at about 2 Gy/day for a total cumulative dose of about 60 Gy) administered concurrently with the alkylating agent temozolomide (TMZ) dosed at 75 mg/m$^2$/day for 42 consecutive days, followed by multiple cycles of adjuvant TMZ (5 days on/23 days off) dose escalated up to 150 mg/m$^2$/day. The modest tumor control achieved with this protocol and the high recurrence rates partly stem from two main limitations: (i) the need to administer an effective total radiation dose in small fractions in order to avoid serious toxicities to fragile brain structures such as the hippocampus, and (ii) the rapid emergence of resistance to TMZ [5].

TMZ has several attractive pharmacological attributes including oral bioavailability, the ability to cross the blood-brain barrier (BBB), and an excellent toxicity profile [6,7]. For practical reasons, the DNA methyl damage inflicted by TMZ can be divided into two main categories: non-O$^6$-methylguanine (of which up to 90% are N-methylpurine adducts) and O$^6$-methylguanine (5–10%) lesions. The former are either N$^7$-methylguanine adducts (which represent the vast majority of lesions generated by TMZ) or N$^3$-methyladenine adducts; all these N-methylpurine lesions are generally but not exclusively repaired by the base excision repair (BER) system which removes and replaces nucleobases damaged by small adducts such as methyl groups [8]. The O$^6$-methylguanine lesions are specifically removed in a stoichiometric reaction by a direct reversal suicide enzyme, the O$^6$-methylguanine DNA methyltransferase (MGMT).

In the clinical practice, when administered to GB patients at dosages corresponding to the Stupp protocol, the peak drug concentration reached by TMZ in the brain is about 10 μM or less [9,10]. At these concentrations, and when MGMT is not present, TMZ kills cancer cells exclusively via O$^6$-methylguanine lesions. This is because, unlike the non-O$^6$-methylguanine lesions, which are rapidly and efficiently repaired by the BER system, the O$^6$-methylguanine ones are irreparable in the absence of MGMT and cytotoxic. According to the prevailing theory, this stems from an irreconcilable conflict between the DNA polymerase which sees the O$^6$-methylguanine chemically as an adenine nucleobase and thus wrongly pairs it with a thymine and the mismatch repair (MMR) system which attempts to correct the problem by removing the wrongly matched base [5,11]. This futile attempt at correcting the O$^6$-MeG-T mispair ultimately leads to apoptosis, a process initiated by the MHS2/MHS6 heterodimer (MutSα) which is believed to be the DNA damage sensing component of the MMR system [8]. In aggregate, TMZ is reliant for its cytotoxicity on the proper function of the MMR repair system, but it becomes ineffective when MGMT is present or when various components of the MMR repair system are compromised (i.e., the state of MMR-deficiency).

Historically, clinical data shows that concurrent TMZ plus radiotherapy is more effective than radiotherapy alone in about 40–45% of GB patients with tumors that exhibit MGMT promoter methylation, but also in about 10–20% of GB patients with MGMT-proficient tumors [12,13]. Unfortunately, these numbers cannot tell much about the ability of TMZ to act as a radiosensitizer because, in order to determine that, data from a group of patients treated with TMZ alone would also be needed. In an effort to clarify this issue, a number of in vitro studies were conducted over the years with mixed results. For instance, some found that the cytotoxic

effects of TMZ plus radiation are only additive [14] while others demonstrated radiosensitization effects [15–17], but not in the MGMT-proficient setting at physiologic concentrations. It is also worth noting however that when the BER system is incapacitated by knocking down its initiating enzyme, clinically relevant concentrations of TMZ can radiosensitize MGMT-proficient GB cells [15]. A more comprehensive animal study was conducted by Carlson et al. [18] which determined that TMZ can indeed act as a radiosensitizer in vivo with the caveat that these effects are rare and only happen in a subset of MGMT-deficient GB xenografts. To summarize, all these limitations of TMZ in the MGMT-proficient setting were indirectly deduced from the original Stupp trial data [19] and subsequent clinical data [20–23], and further validated by animal studies [18]. The MGMT protein remains a formidable obstacle to chemoradiation in ways that are incompletely understood, including the ability of this protein to contribute to radioresistance in a fashion that is independent of its primary role of reversing $O^6$-methylguanine lesions [19].

There are several strategies that can potentially improve on the efficacy of radiotherapy in gliomas which include increasing the precision of dose delivery (e.g., image-guided intensity modulated radiotherapy, brachytherapy, etc.), optimizing fractionation schemes [24], targeting resistant slow proliferating or hypoxic subpopulations (e.g., hypoxic cell radiosensitizers, etc.) by achieving a differential radiosensitization of cancerous tissues while radioprotecting sensitive normal tissues [25], and scheduling standard chemotherapy to be administered together with targeted DNA repair inhibitors (e.g., synthetic lethality) [26,27]. However, despite some promising preclinical outcomes, none of these strategies proved to work very well in the clinic thus far [2]. Due to the highly infiltrative nature of GB and its exquisite adaptation to the highly metabolic microenvironment of the brain tissue, the precision delivery of external-beam radiotherapy (EBRT) into tiny distant tumor foci, even when using computer-assisted stereotactic radiosurgery, remains very challenging. Similarly, hypoxic cell radiosensitizers didn't pan out as they fail to target distant, much better oxygenated, secondary tumor microfoci which tend to hide in normal brain areas. The synthetic lethality strategy holds a much better promise in GB, but it's still too early to tell if this strategy will prove to have a long-lasting impact on the median survival of GB patients [27].

Another tack explored by pharma companies in the past was to improve on the radiosensitization properties of TMZ by further modifying the chemical structure of this alkylating agent [6]. One way to increase the potency and/or the radiosensitization properties of TMZ is by creating so-called TMZ analogs—i.e., TMZ variants in which only the alkyl group that is transferred to DNA is modified. However, most of the gains in potency achieved by these analogs generally happen at the expense of additional toxicities, meaning that a mechanistic link must exists between the chemical nature of the alkyl group that is transferred to DNA and the frequency of off-target toxicities (such as those in the bone marrow). Finally, another strategy is to chemically derivatize TMZ at other positions (other than the alkyl group) in order to generate derivatives with superior tumor uptake and therefore improved potencies. Banking on this latter strategy, we generated a novel chemical entity—called NEO212—according to a manufacturing scheme we have previously published (US patent no. 9,522,918) which entails the derivatization of TMZ with perillyl alcohol, a natural monoterpene with unique solvent properties (reviewed in [28]).

Our previous observations showed that NEO212 is significantly more potent than TMZ over a range of concentrations including the clinically relevant range for gliomas [29], which led us to suspect that NEO212 might be taken up by tumor cells more efficiently than TMZ. If NEO212 has a higher tumor availability, this could lead to an improved alkylation differential in favor of NEO212 at equivalent dosages. Therefore, we advanced the hypothesis that NEO212 might perform significantly better than TMZ at exploiting non-$O^6$-methylguanine

lesions when administered concurrently with radiation. In theory, this enhanced alkylation capacity of NEO212 at clinically relevant concentrations might successfully overwhelm the BER system thus allowing for better synergisms between this drug and radiation irrespective of MGMT status. We further hypothesized that the effects of NEO212 on BER in TMZ-resistant GB cells treated with NEO212 will be aggravated in the presence of minimally cytotoxic concentrations of PARP inhibitors (PARPi). It is well established that the catalytic activity of PARP-1 [30,31] or PARP-2 [32] is required for the stabilization of BER intermediates generated during the repair of methyl damaged purines in a process that ultimately optimizes the downstream homology directed repair (HDR) flux and streamlines the replication repair following genotoxic insults inflicted by DNA methylating agents. Moreover, PARPi were already shown to trap PARP-1/2 at DNA damaged sites in a process that significantly prolongs the half-lives of BER intermediates, delays the completion of BER, and further increases the chance for replication forks to collide with these structures and stall. In turn, these delays make the replication forks increasingly more vulnerable to the effects of ionizing radiation (IR). Therefore, the addition of minimally cytotoxic amounts of PARPi is expected to aggravate the effects of NEO212 on BER and unmask additional synergies.

To test the above hypotheses, we employed gas chromatography/mass spectrometry (GC/MS) and high-performance liquid chromatography (HPLC) analyses to determine the tumor cell uptake of NEO212 relative to TMZ. We further measured the brain and tumor uptake of NEO212 and TMZ after oral administration in non-tumor bearing and tumor bearing animals, while also assessing the bone marrow toxicity of NEO212 in a separate dose escalation study. Finally, we conducted high content in vitro experiments in which chemoradiation synergisms were assessed by quantifying irreparable DNA double strand breaks (DSBs) in GB cell lines, including isogenic variants with different mechanisms of TMZ resistance, treated concurrently with either NEO212 or TMZ and radiation in the presence or absence of minimally cytotoxic concentrations of PARPi.

## Materials and methods

### Reagents

TMZ was purchased from Millipore Sigma (Burlington, MA). NEO212 was synthetized by Norac Pharma (Azusa, CA) and kindly provided by NeOnc Technologies (Los Angeles, CA). Olaparib was purchased from LC Laboratories (Woburn, MA). The monoclonal γH2AX antibody (clone JBW601) was purchased from EMD Millipore (Darmstadt, Germany). A secondary Alexa Fluor 647 (AF647) Fab antibody fragment, the Pacific Blue-labeled Annexin V, an Alexa Fluor 488 NHS ester, and the 4',6 diamidino-2-phenylindole (DAPI) nuclear stain were purchased from ThermoFisher (Waltham, MA). All other reagents, including the MGMT inhibitor $O^6$-benzylguanine or O6BG, were purchased from Millipore Sigma (Burlington, MA). Olaparib was purchased from Selleck Chemicals (Houston, TX). DMSO solutions of alkylating drugs were prepared fresh from powder for each experiment.

### Cells

The LN229, T98G, and U251 human glioma cell lines were obtained from ATCC (Manassas, VA) and maintained according to the culture protocols recommended by ATTC. The LN229TR2 is an MMR-deficient variant of LN229 and was generated by our group [29]. The TMZ-resistant variants of LN229 and U251 (i.e., LN229M and U251M) were generated by infecting parental cells with a lentiviral construct in which the complementary DNA sequence (CDS) of human *MGM*T and the CDS for firefly luciferase were cloned in tandem separated by an IRES sequence and under an EF1α promoter while the CDS for enhanced green

fluorescent protein (EGFP) was cloned in a separate open reading frame (ORF) under a CMV promoter. The GL261 murine glioma cell line was a gift from Dr. Alan Epstein, Keck School of Medicine, University of Southern California. The TMZ-resistant variant of GL261 cells (i.e., GL261M) was generated by transfecting the wildtype cells with a plasmid construct carrying the mouse *Mgmt* CDS under the control of a mouse phosphoglycerate kinase 1 (mPGK) promoter. The transfected GL261 cells were further cultured in increasing concentrations of TMZ to select the GL261M line variant. Both MGMT DNA constructs (i.e., the human MGMT/ffLuc/EGFP-expressing lentivirus and the murine Mgmt-expressing plasmid) were purchased from VectorBuilder (Chicago, IL).

## Gas chromatography/mass spectrometry (GC/MS) and high-performance liquid chromatography (HPLC) analyses

NEO212 behaves like a prodrug of TMZ meaning that it releases intact TMZ as the carbamate bond between perillyl alcohol and TMZ will eventually break in aqueous environments. To measure the amount of TMZ released by NEO212 inside tumor cells, GB cell lines incubated with either TMZ or NEO212 for 60 or 120 minutes were pelleted and lysed in methanol. An identical amount of deuterated leucine ($L$-Leucine-$d_{10}$) was then added to all samples as an internal standard. After being further derivatized with trimethylsilyl (TMS), the samples were then applied to the GC/MS instrument using standard techniques and the resultant TMZ peak was identified and quantified based on the signal generated by the D10 leucine internal standard.

## In vivo biodistribution studies in tumor bearing and non-tumor bearing animals

All animal protocols were approved by the IACUC of University of Southern California and animals maintained according to strict guidelines. For the studies conducted in tumor bearing animals, we established glioma tumors by implanting murine GL261 cells into the brains of C57BL/6 mice. For this orthotopic xenograft model, the cells were implanted 3 mm deep in the right hemisphere of mice brains using stereotactic injections. Briefly, following surgical exposure of the skull and drilling a small-bore hole with a dental drill, GL261 ($10^5$ in 2 μl) murine glioma cells were injected slowly over a three-minute time interval using a Hamilton syringe attached to the stereotaxic frame and followed by a two-minute rest period with the syringe needle left in place to minimize leakage from the injection track. The syringe needle was then slowly removed, the hole covered with beeswax, and the surgical incision closed with 2 stitches (3.0 silk). The tumors were allowed to grow for 14 days before one dose of 50 mg/kg of NEO212 was administered by oral gavage in a suspending vehicle (OraPlus) and animals sacrificed at various time intervals. The same dosage (50 mg/kg) of either TMZ or NEO212 was used in the studies involving non-tumor bearing animals. In-house developed HPLC methods were employed for determining the area under the curve (AUC) of intact NEO212 and its metabolites (i.e., TMZ and AIC) in plasma versus the brain parenchyma (BP) in non-tumor bearing and in the BP of the tumor bearing animals. Ethyl acetate was used to extract the NEO212 or TMZ from plasma, cerebrospinal fluid (CSF), and BP samples, using theophylline as an internal standard. All blood and tissue samples were then diluted in acetonitrile prior to analysis. NEO212 and TMZ in mice dosed with either NEO212 or TMZ by oral gavage were separated (isocratic separation) on a Roc 10 x 4.6mm x 3μm C18 column (Restek, Bellefonte, PA). The isocratic mobile phase consisted of acetonitrile + 0.1% trifluoroacetic acid (TFA): water + 0.1% TFA (pH 4.0) (40:60 v/v) in a positive-ion multiple reaction monitoring

mode. The method was validated over the range from 5–2000 ng/mL in mouse plasma, CSF, and BP with respect to linearity, accuracy, precision, selectivity, and stability.

## Functional analysis to confirm MGMT expression and activity by TMZ-resistant cells

Lysates prepared from the 3 isogenic pairs of GB cells (LN229/LN229M, U251/U251M, and GL261/GL261M) and the MMR-deficient LN229TR2 and MGMT-positive T98G cells were analyzed by Western-blotting for MGMT expression levels. A rabbit polyclonal antibody that crossreacts with both human MGMT and mouse Mgmt proteins was purchased from Boster Biological Technology (Pleasanton, CA) and used in this Western-blot analysis. This antibody was raised against a synthetic peptide PVFQQESFTRQVLWK, which corresponds to a sequence in the middle region of human MGMT which is different from the related rat and mouse sequences by only one amino acid. Furthermore, to check the functionality of the MGMT protein in LN229M and U251M cells (infected with a human MGMT/ffLuc/EGFP-expressing lentivirus) and in GL261M cells (transfected with a murine Mgmt-expressing plasmid), we completed a colony formation assay (CFA) with these cells seeded in 6-well plates at a density of 50 cells/cm$^2$ and incubated with a range of TMZ or NEO212 concentrations in the presence or absence of the MGMT inhibitor $O^6$-benzylguanine (Millipore Sigma, Burlington, MA) which was added to a final concentration of 40 μM.

## Analysis of clonogenic survival of GB cells treated with chemotherapy and radiation

The clonogenic survival was performed to determine the effect of clinically relevant concentrations (i.e., 10 μM or less) of TMZ, NEO212 or decayed NEO212 (dNEO212) administered either alone or in combination with radiation (2 Gy) to the long-term survival of a panel of glioma cell lines. Decayed NEO212 was generated after pre-incubating NEO212 for 24 hours in complete Dulbecco's Modified Eagle's Medium (DMEM) before being added the cells. Cells were seeded in 6-well plates at a density of 50 cells/cm$^2$ and allowed to adhere and grow in complete medium for 24 hrs. At the end of incubation period, the cells were treated with a range of TMZ or NEO212 concentrations (from 2 to 10 μM) either alone or concurrently with IR (2 Gy) for 5 consecutive days. All drug treatments were applied 1 hour before irradiation. External beam radiotherapy was administered in an X-RAD320 irradiator (Precision X-Ray, North Branford, CT). After treatments, the cells were allowed to develop colonies over 14 days. The colonies were stained with crystal violet and subsequently counted. Survival was determined as a ratio of plating efficiencies for each irradiated group to that of the unirradiated control. In a separate set of clonogenic survival analyses, the cells were seeded in 6-well plates at a density of 50 cells/cm$^2$ and incubated with a range of concentrations (0-1000nM) of Olaparib with or without radiation therapy.

## Analysis of DNA damage by γH2AX staining

For persistent DSB foci quantification by γH2AX staining, the cells were seeded in 6-well plates at a density of 50,000 cells/cm$^2$ and exposed to 5 consecutive days of treatments with either chemotherapy alone (TMZ or NEO212, 10μM) or IR alone (2 Gy) or concurrent chemotherapy and IR. At 24 hours after the last treatment, the cells were fixed in 4% paraformaldehyde in PBS, permeabilized in 0.05% Triton X-100 in PBS, incubated for 60 minutes at 37˚C with a γH2AX antibody, clone JBW301 (EMD Millipore, Darmstadt, Germany), then washed in PBS and further incubated for another 30 minutes with an AF647-labeled Fab secondary antibody fragment (ThermoFisher, Waltham, MA), and finally counterstained with 4',6 diamidino-

2-phenylindole (DAPI). The plates were imaged in an ImageXpress Micro XLS (Molecular Devices, San Jose) high content widefield microscope system. High-content data was captured from each plate at 10x magnification, which allows for the extraction of up to 81 fields per well. The DSB foci and DAPI-stained areas in each micrograph were quantified digitally by pixel counting on images taken from hundreds of fields per condition using the 'SimplePCI' imaging software (Hamamatsu Corporation, Sewickley, PA). The number of pixels corresponding to the total number of persistent DNA DSB foci were quantified in all fields taken at 10x magnification and expressed as a ratio to the total number of pixels corresponding to DAPI-stained nuclei using the formula 'total DSB pixel counts/total nuclear pixel counts' for each treatment group. These numbers were further normalized to the total number of DSB foci found in the untreated control wells.

## FACS analysis of glioma cell lines

To correlate the amount of cell death with DNA damage (i.e., DSB foci), glioma cells were seeded in 6-well plates at a density of 50,000 cells/cm$^2$ and exposed to 5 consecutive days of treatments with either chemotherapy alone (TMZ or NEO212, 10 μM) or IR alone (2 Gy) or concurrent chemotherapy and IR. The medium was replenished every day for 5 consecutive days. At 24 hours after the last treatment, the cells were harvested, resuspended in Annexin V buffer and stained with Pacific Blue-labeled Annexin V for 30 minutes at 37˚C, then pelleted and resuspended in a solution of 4% paraformaldehyde in Annexin V buffer. The cells were fixed in paraformaldehyde for 10 minutes, then pelleted and permeabilized in 0.05% Triton X-100 in Annexin V buffer for 10 minutes, washed in Annexin V buffer, and then incubated for 60 minutes at 37˚C with an AF488-labeled γH2AX antibody, clone JBW301 (EMD Millipore, Darmstadt, Germany), before being finally washed in Annexin V buffer one last time. The cells were further analyzed on a BD FACSAria II instrument (BD Biosciences, Franklin Lakes, NJ) equipped with violet and blue lasers. Cells incubated with an isotype control antibody (i.e., negative control) were used for setting cytometer voltages.

## Bone marrow toxicity studies with NEO212

Groups of mice (n = 10) were dosed orally over 2 weeks with either NEO212 in OraPlus (i.e., the treatment groups) or OraPlus only (i.e., the control group) using a scheme of administration of 5 days on/2 days off. NEO212 in OraPlus was given as either 50 mg/kg/day or 120 mg/kg/day or 150 mg/kg/day. The weights of the animals were monitored for 15 days in total. At the end of the study the mice were euthanized and the femurs from all animal groups sectioned and stained with H&E to assess the changes in bone marrow morphology and cellularity.

## Statistical analysis

Statistical significance was analyzed in Prism v.8.2.1 (GraphPad Software, La Jolla, CA) by unpaired Student's t-test followed by the determination of F values to compare variances. The cell viability and immunocytochemistry data were assessed by analysis of variance (ANOVA) with a significant overall F-test followed by Tukey post-hoc multiple comparison tests of treatment groups relative to control. Two-tailed $P<0.05$ were considered significant.

## Results

### NEO212 shows superior in vitro and in vivo tumor uptake compared to TMZ at physiologic concentrations

The chemical structures of NEO212 and TMZ are shown in **Fig 1**.

**Fig 1. The chemical structures of TMZ and NEO212.** TMZ was conjugated with perillyl alcohol via a carbamate bridge to generate NEO212. The derivatization of TMZ with perillyl alcohol creates a new chemical entity with new physicochemical properties but presumably retains the same alkylating properties of TMZ. This is because NEO212 breaks down to release intact TMZ and therefore the same methyl group (in red) in NEO212 as in TMZ will eventually be donated by NEO212 to DNA nucleobases via the release of the same highly reactive methanediazonium chemical species.

To assess the relative tumor cell uptake of these drugs, we first conducted GC/MS analyses on TMZ-sensitive and TMZ-resistant cells (**Fig 2**, **Panel A**) incubated with either NEO212 or TMZ. The results from these analyses confirm that NEO212 breaks down into TMZ after

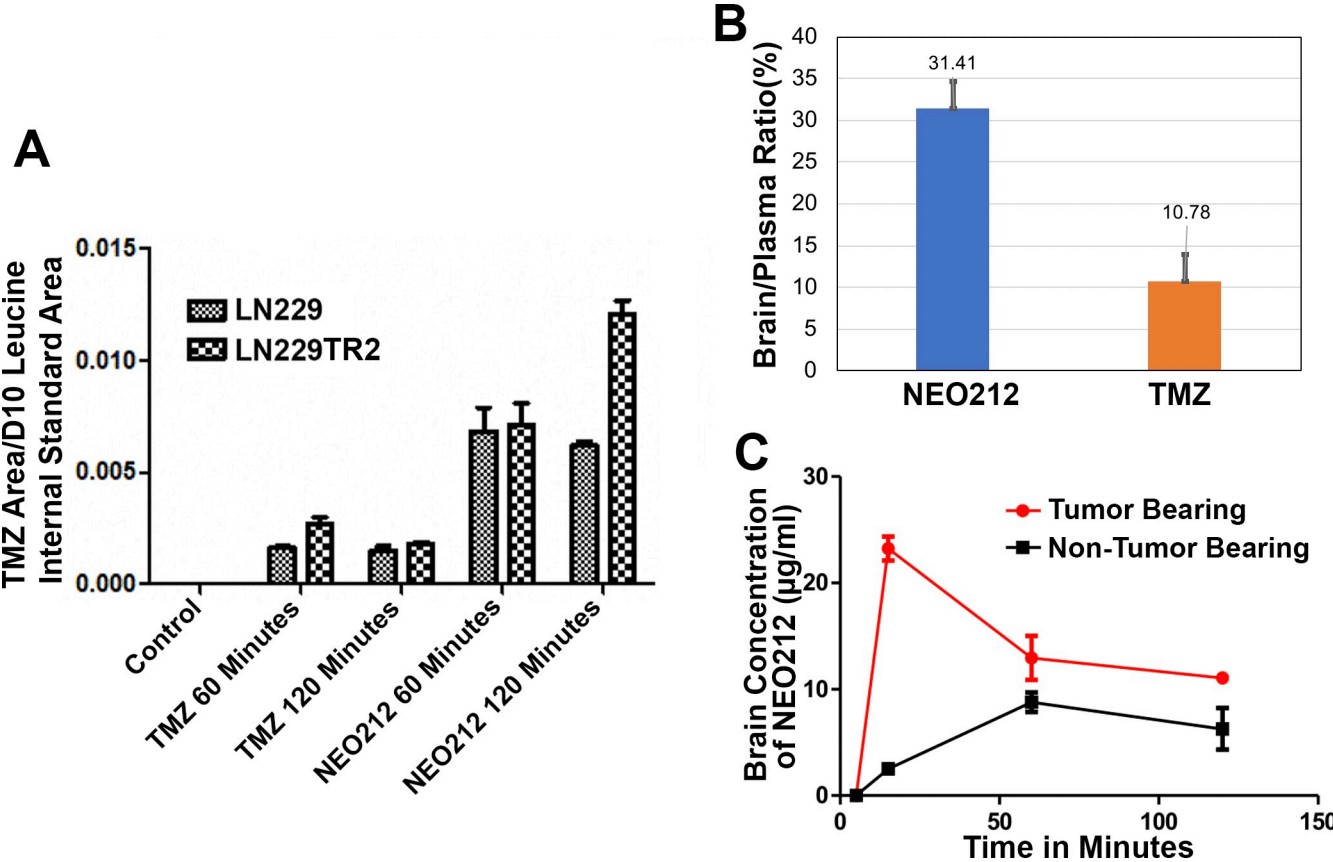

**Fig 2. NEO212 acts as a prodrug of TMZ and has a superior tumor uptake profile.** Based on GC/MS analyses, NEO212, once taken up by tumor cells, appears to break down into intact TMZ and perillyl alcohol. The same analyses show that at equivalent dosages a larger amount of TMZ is released from NEO212 inside GB cells compared to the amount of TMZ taken up by cells after the cells were incubated with equimolar concentrations of either NEO212 or TMZ (Panel A). Brain biodistribution data with NEO212 and TMZ indicate that NEO212 has a higher availability into the brain than TMZ after oral administration of the drugs presumably due to a better penetration of NEO212 through the blood-brain barrier (Panel B). NEO212 also shows a favorable differential uptake (i.e. higher uptake by the diseased brain) when administered orally to brain tumor bearing vs. non-tumor bearing mice (Panel C). For the in vivo studies, the drugs were measured in tissue homogenates by HPLC methods.

being taken up by cancer cells. The data also show that the amount of TMZ released intracellularly by NEO212 after the cells were incubated with equimolar concentrations of either drug is much higher than the amount of TMZ taken up by the cells directly from the medium. Moreover, by using an HPLC protocol developed in house we measured the brain/plasma ratios of TMZ and NEO212 in non-tumor bearing animals. The animals were dosed with 50 mg/kg of either TMZ or NEO212 administered by oral gavage in a suspending vehicle (OraPlus) and euthanized 1 hour later when the blood and the brains were collected, and the tissues further prepared for HPLC analysis. The HPLC data show (**Fig 2, Panel B**) that NEO212 has a much better penetration into the brain than TMZ. Finally, by employing the same HPLC methods, we measured in the GL261 murine GB model, with the tumor cells implanted stereotactically into the brain, the differential uptake of NEO212 by the diseased brain vs. the normal brain. We found that the uptake of NEO212 by the tumoral brain is much higher than the uptake of the drug by the normal brain and it peaks at about 20 minutes after the oral administration of the drug (**Fig 2, Panel C**).

## NEO212 as monotherapy is more potent than TMZ when tested on isogenic pairs of GB cell lines

The efficacy of NEO212 as monotherapy had been previously validated in vitro against multiple TMZ-resistant GB cell lines including glioma stem cells [33], as well as in a xenograft model of TMZ-resistant GB [29]. Building upon these initial studies, we decided to further test NEO212 using a more clinically relevant in vitro approach which was designed to mimic the concurrent component of the Stupp protocol in terms of schedules of administration and dosages. To accomplish this, we first prepared MGMT-expressing versions of two well-studied TMZ-sensitive cell lines (i.e., the human LN229 and U251) by infecting them with a lentivirus that stably expresses human MGMT as well as two reporter genes (EGFP and firefly luciferase) (**S1 Fig**). We additionally generated a murine Mgmt-expressing version of the mouse GL261 cell line (**S1 Fig**), and also included an MMR-deficient version of human LN229 (called LN229TR2) [29] and the human T98G cell line (which constitutively expresses MGMT) in our studies.

After a selection step (i.e., by FACS sorting in the case of human GB lines or by serial passaging in TMZ-containing medium in the case of murine GB cells), we then confirmed the expression of MGMT by all these cells by Western blotting, which validated the successful expression of the artificially introduced MGMT in LN229M, U251M, and GL261M cells (**Fig 3**, **Panel A**). Lastly, we determined the relative potencies of both TMZ and NEO212 drugs against these isogenic pairs of GB cells in a colony formation assay (CFA) in the presence or absence of the MGMT inhibitor O6BG ($O^6$-benzylguanine) and confirmed that: (i) NEO212 kills MGMT-negative GB cells and their MGMT-positive isogenic variants with relatively higher potency than TMZ (**Fig 3**, **Panel B**), and (ii) the artificially introduced MGMT protein was indeed functionally active as it responded to specific O6BG inhibition.

## NEO212 synergizes with IR in the TMZ-resistant setting

In order to test the ability of NEO212 to synergize with IR in a scenario that mimics the Stupp protocol, we seeded TMZ-resistant GB cells at very low densities (50 cells/cm$^2$) in 6-well plates and then treated them once daily for 5 consecutive days with clinically relevant concentrations of either TMZ or NEO212 alone, radiation alone, or as combined treatments. The clonogenic survival data (**Fig 4**, **S2 Fig**) demonstrate that NEO212 can indeed radiosensitize these isogenic pairs of GB cell lines in a concentration range that is clinically relevant (i.e., 2–10 μM). The data further suggest that TMZ can also reach intracellular levels of DNA alkylation that are high enough for synergistic effects to take place with radiation, but only at supra-physiological concentrations such as 20 μM, a value that is not achievable with TMZ in brain in the clinic (**Fig 4**, **S2 Fig**).

Collectively, these CFA data suggest that, unlike TMZ, NEO212 appears to reach optimal intracellular levels of DNA alkylation at much lower concentrations (i.e., 2–10 μM), which may explain its consistent radiosensitization effects in this physiologic range. These effects may not be however exclusively dependent on the alkylating capacity of NEO212 (i.e., the ability of the molecule to break down in the aqueous cytosolic compartment and release the highly reactive methanediazonium, also known as the methyldiazonium ion, which further transfers methyl groups to DNA nucleobases). It may very well be that additional, non-alkylation-related mechanisms, are also contributing to this effect. To further test this hypothesis, we repeated our CFA analyses with decayed NEO212 (dNEO212): i.e., NEO212 pre-incubated in DMEM medium for 24 hours before being added to the cells (**S3 and S4 Figs**). It is well established that in aqueous solutions and in the physiologic pH range, TMZ undergoes a hydrolytic ring opening and breaks down into MTIC or 5-(3-methlytriaz-1-en-1-yl)-1H-imidazole-

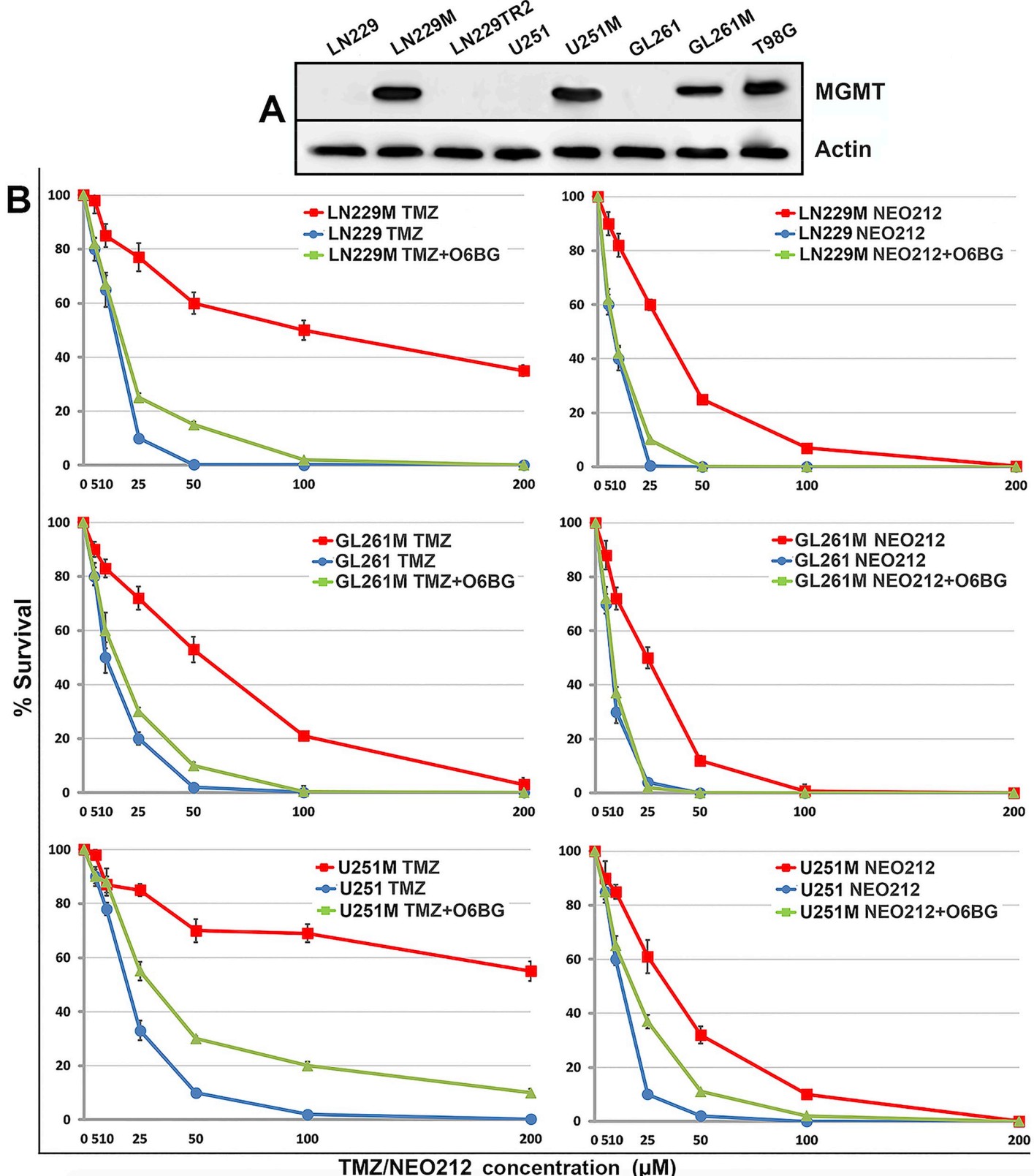

**Fig 3. NEO212 is more potent than TMZ over a broad range of concentrations.** MGMT-expressing variants (i.e., LN229M, U251M, and GL261M) of three well-established GB cell lines were checked along with their parental cells as well as the T98G (an endogenous high MGMT expressor) and LN229TR2 (an MMR-deficient

variant of LN229) GB cells for MGMT expression by Western blotting (panel A). Colony survival data are also shown for all three isogenic GB cell lines (panel B). The O6BG MGMT inhibitor (40 μM) was used to demonstrate that the artificially expressed MGMT proteins in LN229M, U251M, and GL261M cells were functional and respond to specific inhibition (i.e., the MGMT-positive cells are sensitized to both alkylating agents after the addition of the O6BG inhibitor).

4-carboxamide which is a short-lived chemical species that further releases the highly reactive methyldiazonium ion [6]. This known lack of stability of imidazoterazines in aqueous solutions and physiologic pH-es means that after incubating a 10 μM solution of TMZ in DMEM for 24 hours at 37°C, the drug will completely lose its DNA alkylation capacity (i.e., the entire amount of drug will be degraded to AIC or 4-amino-5-imidazole-carboxamide and all of the released methanediazonium will be spent). Because NEO212 is a prodrug of TMZ, it behaves the exact same way as TMZ does when incubated in a medium like DMEM for 24 hours at

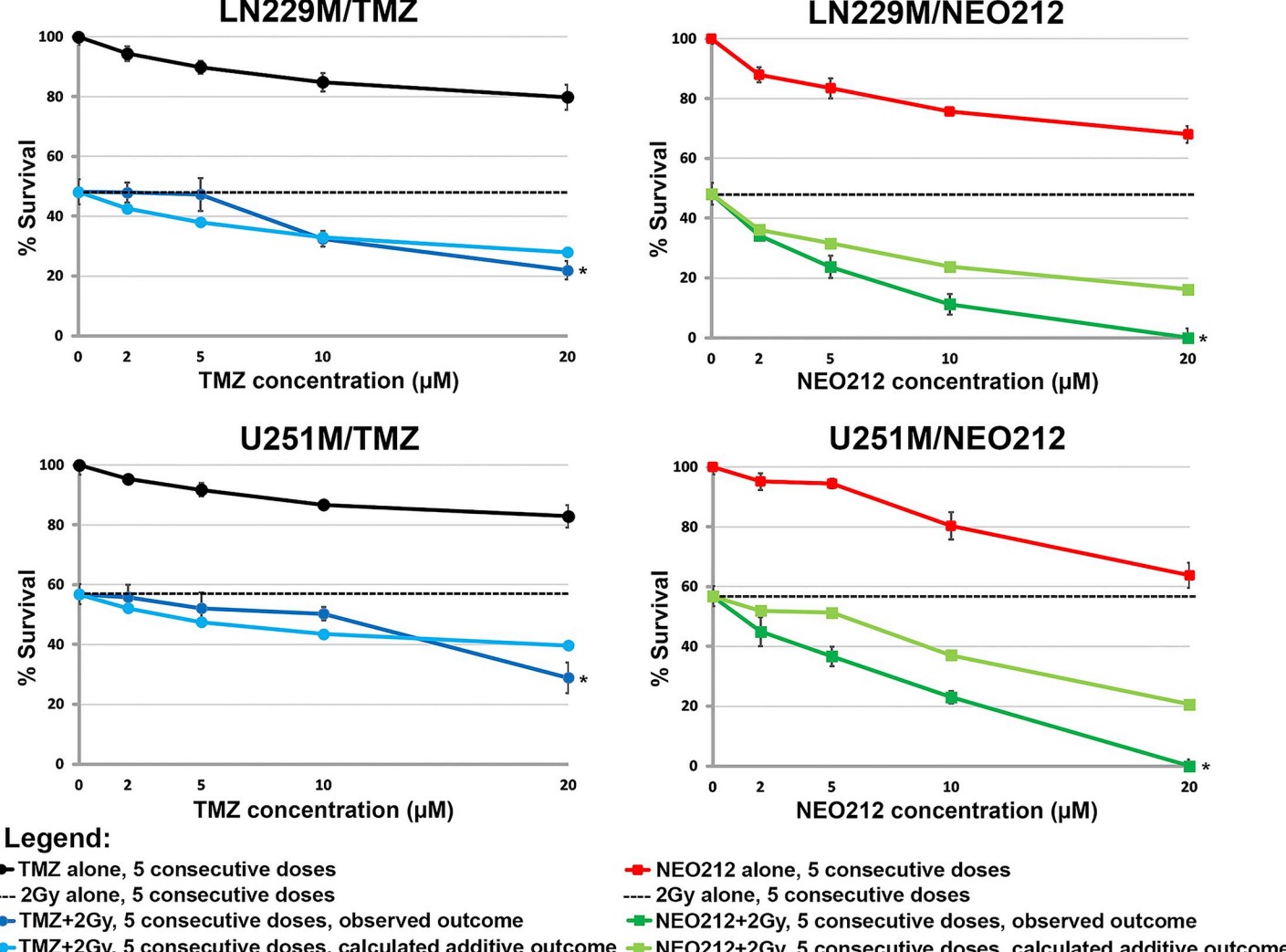

**Fig 4. NEO212 can radiosensitize TMZ-resistant GB cells at clinically relevant concentrations.** In these CFA analyses, we tested both TMZ and NEO212 drugs using a treatment schedule designed to mimic the Stupp protocol which allows for synergisms to be observed (i.e., five consecutive days of concurrent chemotherapy plus radiotherapy vs. five consecutive days of monotherapies). The colony survival data show that NEO212 can synergize with ionizing radiation in the clinically relevant concentration range (i.e., 10 μM or less), whereas TMZ becomes synergistic only outside (i.e., >10 μM) this concentration range where presumably achieves levels of DNA alkylation that are optimal for synergistic effects to take place (* indicates a p<0.01 determined by Student's t-test).

37˚C (i.e., the entire amount of NEO212 will be decayed at the end of this incubation interval). Our clonogenic survival data indicate that when dNEO212 is used instead of the intact molecule, the entire cytotoxic activity of the drug (and its ability to synergize with radiation) is lost (**S3** and **S4** Figs). Therefore, the cytotoxic and radiosensitization properties of NEO212 are exclusively dependent on its alkylation capacity while the perillyl alcohol moiety of NEO212 does not appear to contribute in any way to its ability to function as a radiosensitizer.

## Quantitative immunocytochemistry (qICC) data confirm that NEO212 synergizes with IR in the TMZ-resistant setting and this effect is further enhanced by PARP inhibition

By employing an ImageXpress Micro XLS (Molecular Devices, San Jose) high content wide-field microscope system that is capable of extracting data-dense ICC information from microtiter plates, we were able to generate highly quantitative measurements of the DNA double strand breaks inflicted by both TMZ and NEO212 in combination with radiation to GB cell lines. This highly quantitative measuring of irreparable DNA double strand breaks (i.e., defined as pan-nuclear γH2AX staining persisting at 24 hours after the last treatment [34]) on several thousands of images extracted from 96-well plates allowed us to conduct a more precise analysis of increased granularity of the radiosensitization effects of NEO212. Accordingly, to better define the radiosensitization effects of TMZ or NEO212 at molecular level, we seeded GB cells in 96-well plates at more physiologically relevant densities (i.e., 50,000 cells/cm$^2$) before subjecting them to either TMZ or NEO212 alone, radiation alone, or combination modalities. These treatments were administrated using the same schedule (5 consecutive days of treatment) previously employed in our CFA setup. Our readout of irreparable DNA damage inflicted by the treatments was the formation of persistent γH2AX foci, whereby γH2AX is defined as a phosphoserine residue in the H2AX histone that serves as marker double strand breaks (DSBs) in DNA [35]. The studies of the dynamics of γH2AX formation and resolution after genotoxic therapies have shown that these foci usually completely resolve (i.e., disappear) at 24 hours after the application of the damaging genotoxic insult, a phenomenon which essentially indicates the completion of DNA repair process [36]. If persistent beyond 24 hours, the lasting presence of γH2AX foci was shown to correlate well with irreparable DNA damage [36]. Therefore, in our experimental approach, we applied 5 consecutive days of treatments after which we waited for another 24 hours to allow for all γH2AX foci associated with successful DNA repair to resolve before we fixed and stained the plates with a γH2AX antibody. This approach ensured that only the irreparable DNA foci inflicted by the treatments were actually captured and quantified. Our results show that, even at much higher cellular densities, NEO212 is significantly more potent than TMZ (**Figs 5 and 6, S5** and **S6** Figs) based on the extent to which this drug leads to the development of persistent γH2AX foci as revealed by qICC analyses. Importantly, this difference in potency that favors NEO212 over TMZ appears to also hold true in the MGMT-proficient or MMR-deficient settings.

In order to lend further credence to our hypothesis that NEO212 can, in the lower micro-molar range, overwhelm the BER repair system (i.e., the system responsible for the repair of up to 90% of DNA lesions inflicted by the drug), we decided to also investigate the outcome of combining NEO212 or TMZ with minimal cytotoxic amounts (10 nM) of Olaparib, a PARPi. Before testing any combinations, we first examined the cytotoxicity of Olaparib as a monotherapy to GB cells lines in CFA analyses (**S7 Fig**).

The PARylation of DNA repair proteins represents a critical event in at least three distinct DNA repair pathways: the BER repair, the homologous recombination (HR) repair, and a

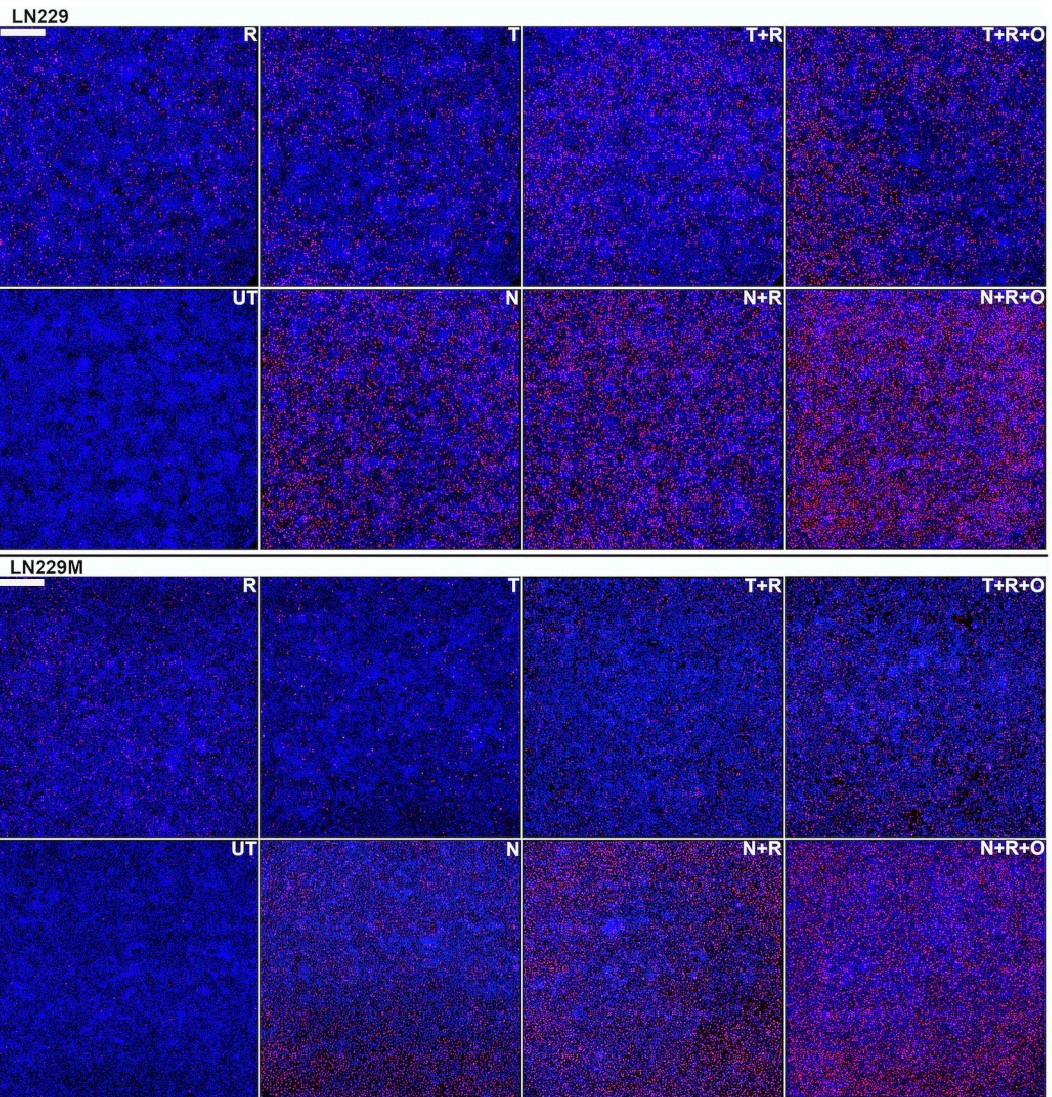

**Fig 5. Quantitative ICC analysis of irreparable DNA damage in LN229/LN229M GB cells.** Cells were seeded at high densities (50,000cells/cm$^2$) and either left untreated (UT) or treated for five consecutive days with either 10 μM TMZ (T) or NEO212 (N) or 2 Gy (R) alone or combinations without (T+R or N+R) or with (T+R+O or N+R+O) Olaparib (O). The cells were probed with a γH2AX antibody and an AF647-labeled secondary and nuclei were counterstained with DAPI. Persistent γH2AX foci (red) were digitally counted relative to the total number of cell nuclei (blue). Each panel is data dense and represents a composite of 36 fields in total (i.e., a square of about 3x3 mm) captured on a widefield microscopy instrument and digitally stitched together. Scale bar is 500 μm (upper left corner).

variety of alternative end joining repair called microhomology-mediated end joining (MMEJ) [34,37]. The last two function as critical DNA repair mechanisms of DSBs resulting from radiation damage during the S and G2 phases of the cell cycle. Therefore, the possibility that Olaparib might also act as a radiosensitizer in its own right at minimally cytotoxic concentrations had to be ruled out first by treating the cells with the Olaparib/IR combination in the absence of alkylating agents. Although Olaparib was reported in the literature to be a radiosensitizer at higher concentrations, we found that a concentration of 10 nM did not sensitize any of the GB cell lines tested to 2 Gy of IR (**S8 Fig**).

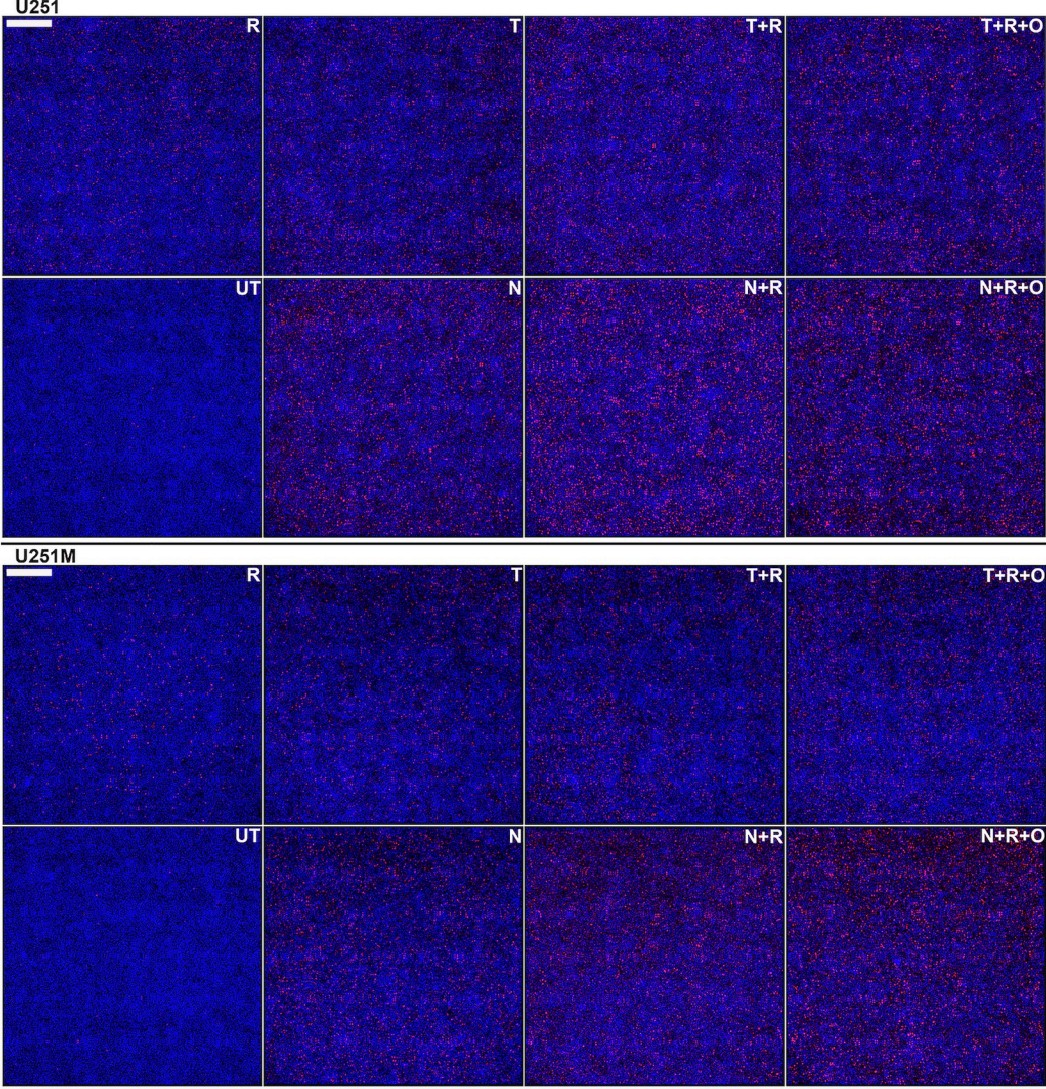

**Fig 6. Quantitative ICC analysis of irreparable DNA damage in U251/U251M GB cells.** Cells were seeded at high densities (50,000 cells/cm$^2$) and either left untreated (UT) or treated for five consecutive days with either 10 μM TMZ (T) or NEO212 (N) or 2 Gy (R) alone or combinations without (T+R or N+R) or with (T+R+O or N+R+O) Olaparib (O). The cells were probed with a γH2AX antibody and an AF647-labeled secondary and nuclei were counterstained with DAPI. Persistent γH2AX foci (red) were digitally counted relative to the total number of cell nuclei (blue). Each panel is data dense and represents a composite of 36 fields in total (i.e., a square of about 3x3 mm) captured on a widefield microscopy instrument and digitally stitched together. Scale bar is 500 μm (upper left corner).

As predicted by our hypothesis, we were able to unravel additional synergisms with PARPi following the addition of minimally cytotoxic amounts of Olaparib to concurrent NEO212 or TMZ and radiation (**Figs 5 and 6, S5 and S6 Figs**).

## The quantification of DSBs shows that NEO212 causes more irreparable DNA damage than TMZ

The DNA damage induced to isogenic pairs of GB cells by combinatorial treatments was quantified from the qICC data collected from our widefield microscopy analyses of persistent DSB foci (using γH2AX as a biomarker of DSB DNA damage). The quantification (by pixel

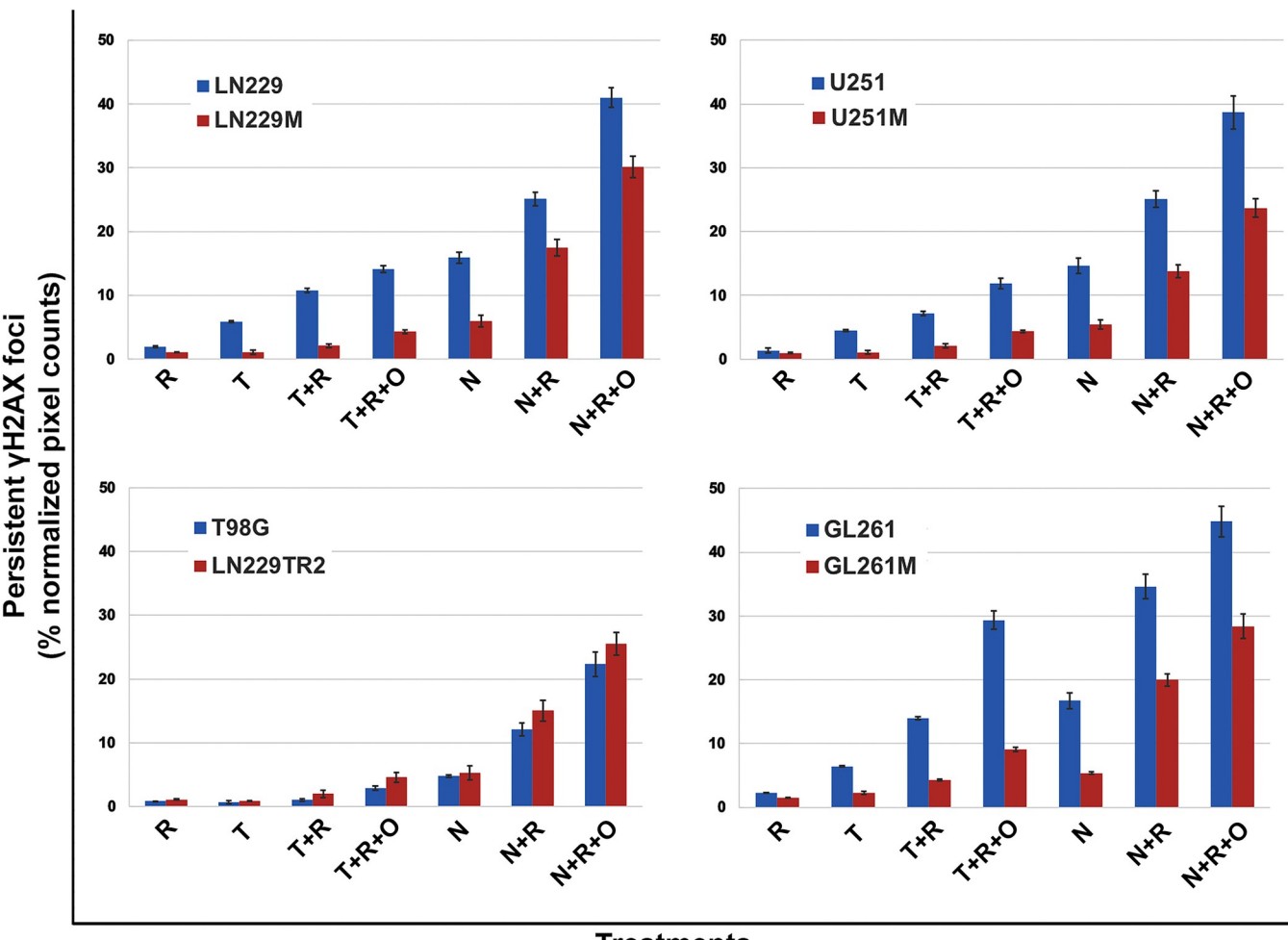

**Fig 7. A quantification of qICC γH2AX data of irreparable DNA damage in GB cells.** Cells were seeded at high densities (50,000 cells/cm$^2$) and either left untreated or treated for five consecutive days with either 10 µM TMZ (T) or NEO212 (N) or 2 Gy (R) alone or combinations without (T+R or N+R) or with (T +R+O or N+R+O) Olaparib (O). Persistent γH2AX foci were digitally counted and expressed relative to the total number of cell nuclei. All ratio values (pixel counts from treatments expressed as a % ratio of persistent γH2AX foci to nuclei values normalized to untreated counts) were found highly statistically significant with p values of <0.001 (ANOVA with Tukey post-hoc testing).

counting of γH2AX stained areas vs. nuclear staining) of treatment fields captured by widefield microscopy for all single and combinatorial treatments revealed the exact same trends for TMZ and NEO212 as previously seen with the data obtained from CFA analyses. These qICC data confirm that, even when cells are seeded at higher cellular densities, NEO212 is still significantly more potent (based on the extent of irreparable DNA damage inflicted by this molecule) than TMZ (**Fig 7**).

## The amount of irreparable DNA damage inflicted by NEO212 correlates well with cell death

To better understand whether the DNA damage inflicted by NEO212 correlates with cell death, we employed the same experimental approach used for qICC by seeding the cells again at high densities (i.e., 50,000 cells/cm$^2$) in 6-well plates and treating them daily for five consecutive days. At 24 hours after the last treatment, we harvested the cells, stained them with

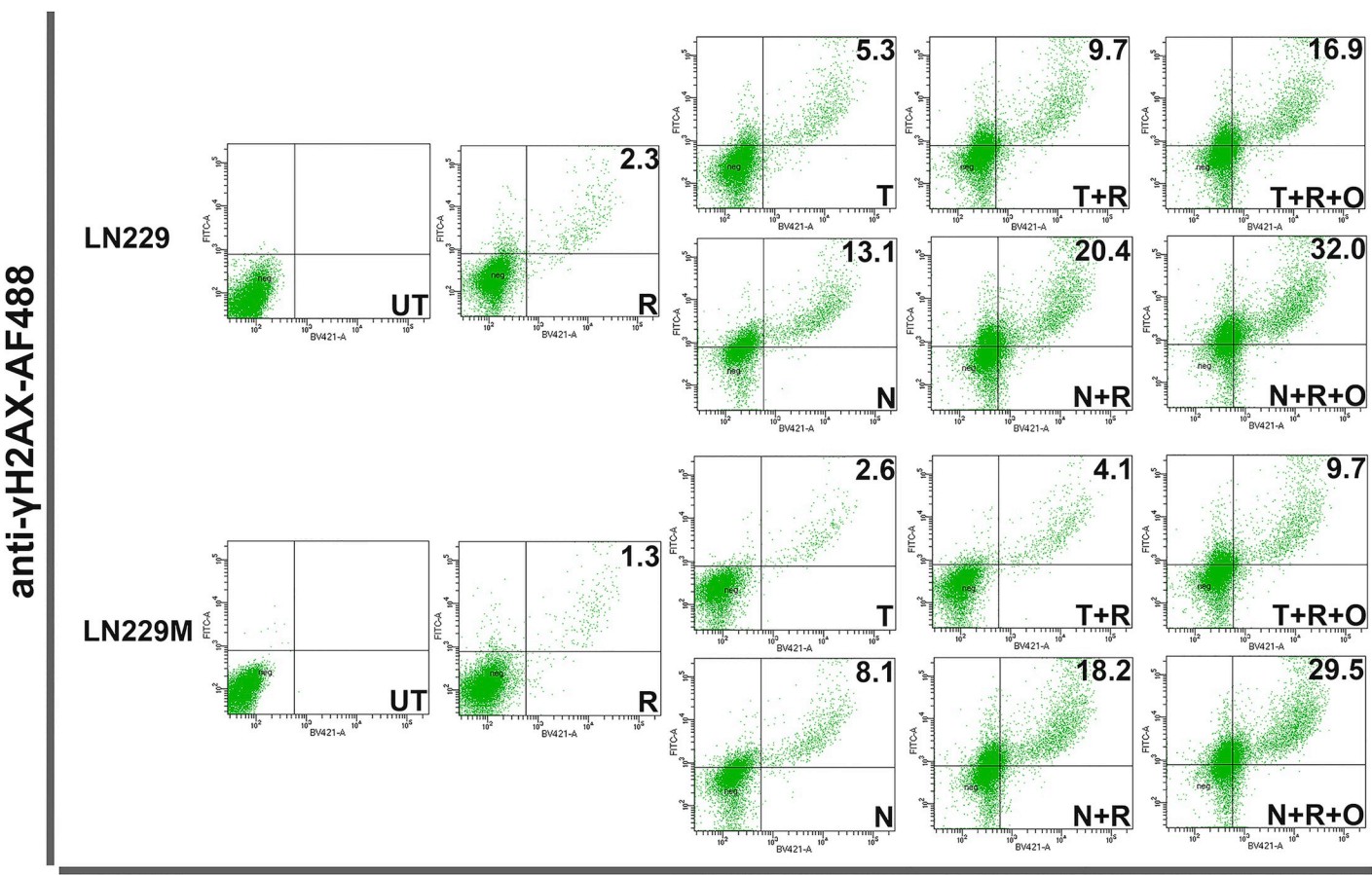

**Fig 8. FACS analysis of cell death resulting from irreparable DNA damage.** LN229 and LN229M cells were seeded at high densities (50,000 cells/cm²) and were either left untreated (UT) or treated for five consecutive days with either 10µM TMZ (T) or NEO212 (N) or 2 Gy (R) alone or combinations without (T+R or N+R) or with (T +R+O or N+R+O) Olaparib (O). The cells were probed with a Pacific Blue-labeled Annexin V and then fixed, permeabilized and probed with an AF488-labeled γH2AX antibody. γH2AX/Annexin V double positive cells (i.e., dead cells due to irreparable DNA damage) are shown as percentages of total cell numbers. Representative panels are shown from three independent experiments.

Pacific Blue-Annexin V, and then further fixed, permeabilized, and stained them with an AF488-labeled γH2AX antibody before FACS analyzing them. The FACS data (**Figs 8 and 9, S9 and S10 Figs**) confirmed the same trends we previously observed with our CFA and quantitative ICC analyses for both TMZ and NEO212. In addition, the FACS experimental setup allows for drawing direct correlations between the amount of persistent irreparable DNA damage (i.e., the γH2AX signal) and the amount of cell death (i.e., the externalization of phosphatidylserine signal) inflicted by NEO212 or TMZ.

## Calculation of radiosensitization potency of NEO212

Based on the above experimental observations, we then calculated the fold radiosensitization values for both TMZ and NEO212 using the same formula for all datasets (i.e., the ratio between the total cytotoxic effect observed with combinatorial treatments divided by the calculated additive effects of treatments as monotherapies). The same formula was also used when we calculated the fold sensitization values for the radiosensitization activity of minimally cytotoxic concentrations of Olaparib (**S8 Fig**). The tabulation of calculated fold radiosensitization

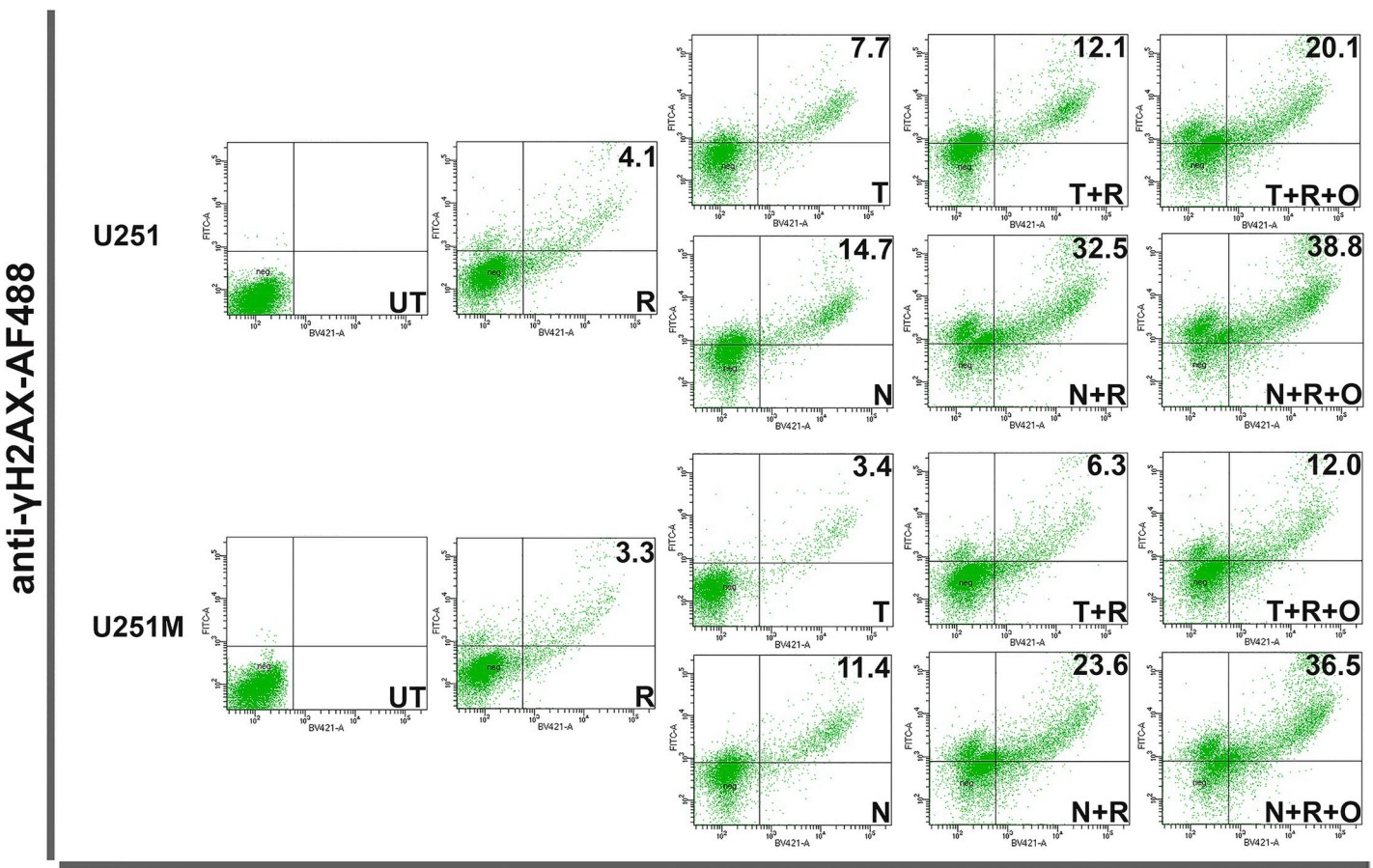

**Fig 9. FACS analysis of cell death resulting from irreparable DNA damage.** U251 and U251M cells were seeded at high densities (50,000 cells/cm$^2$) and were either left untreated (UT) or treated for five consecutive days with either 10 μM TMZ (T) or NEO212 (N) or 2 Gy (R) alone or combinations without (T+R or N+R) or with (T+R+O or N+R+O) Olaparib (O). The cells were probed with a Pacific Blue-labeled Annexin V and then fixed, permeabilized and probed with an AF488-labeled γH2AX antibody. γH2AX/Annexin V double positive cells (i.e., dead cells due to irreparable DNA damage) are shown as percentages of total cell numbers. Representative panels are shown from three independent experiments.

values from our qICC and FACS analyses (**Fig 10, Panel A**) strongly indicate that NEO212 performs significantly better as a radiosensitizer than TMZ. This discrepancy in the radiosensitization characteristics of the two molecules becomes readily apparent particularly in the TMZ-resistant setting of either MGMT-proficiency or MMR-deficiency (**Fig 10, Panels B and C**).

The addition of very low (i.e., minimally cytotoxic) concentrations of a PARP inhibitor, with the intention to further incapacitate the BER repair system involved in the repair of non-O$^6$-methyguanine lesions, was shown to further enhance the radiosensitization effects of NEO212 even more. However, it is noteworthy that while these synergisms appear significantly more pronounced with NEO212 in the presence of BER inhibition, these effects also become apparent (albeit less pronounced) with TMZ in the same physiologic concentration range. These outcomes suggest that NEO212 in combination with IR can better exploit the non-O$^6$-methyguanine lesions created by NEO212 which lead to robust synergistic effects irrespective of the mechanism of resistance (i.e., either MGMT overexpression or MMR-deficiency). In aggregate, all these observations suggest a possible mechanistic explanation for the

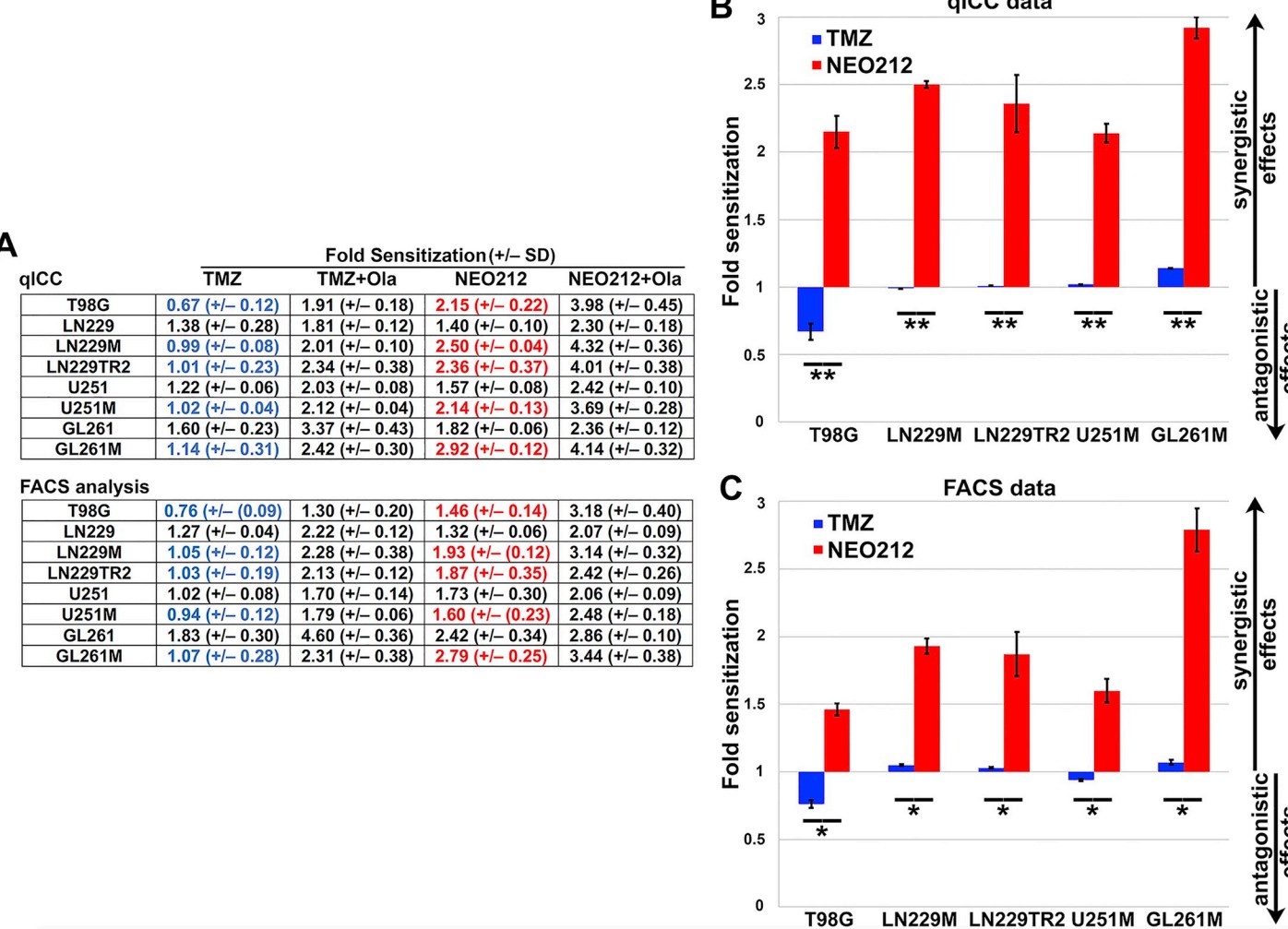

**Fig 10. NEO212 is a robust radiosensitizer of TMZ-resistant GB cell lines.** Fold sensitization values were determined from qICC and FACS datasets. Values >1 indicate synergistic effects (i.e., radiosensitization). Values equal to 1 indicate additive effects and values <1 indicate antagonistic effects. Panel A shows the tabulated fold sensitization data for TMZ and NEO212 either as monotherapies or in combination with Olaparib (Ola). While TMZ cannot sensitize MGMT-proficient or MMR-deficient cell lines to ionizing radiation (values in blue), NEO212 appears to consistently do so (values in red). The blue (TMZ) and red (NEO212) values for TMZ-resistant cell lines are graphed in Panels B and C. A low concentration of Olaparib (i.e., 10 nM) added to either TMZ or NEO212 further improve the radiosensitization profiles of both drugs. All values were found statistically significant ($^*$signifies a $p<0.01$ while $^{**}$signifies a $p<0.001$ as determined by ANOVA with Tukey post-hoc testing).

radiosensitization properties of NEO212. Accordingly, the concurrent administration of NEO212 and IR at physiologic dosages might mimic a synthetic lethality scenario (due to the exhaustion of BER repair mechanism incited by NEO212) which may significantly further benefit from the addition of very low doses of a PARP inhibitor.

## A dose escalation in vivo toxicity study shows that NEO212 is well tolerated at clinically relevant dosages

To assess the toxicity of NEO212 in vivo, we conducted a preliminary in vivo study in which we analyzed the bone marrows of non-tumor bearing C57BL/6 mice in which NEO212 was dose escalated up to 150 mg/kg/day by oral gavage over a short course of 2 weeks of treatment (**Fig 11**).

| Vehicle | NEO212 120mg/kg/day | NEO212 150mg/kg/day |
|---------|---------------------|---------------------|

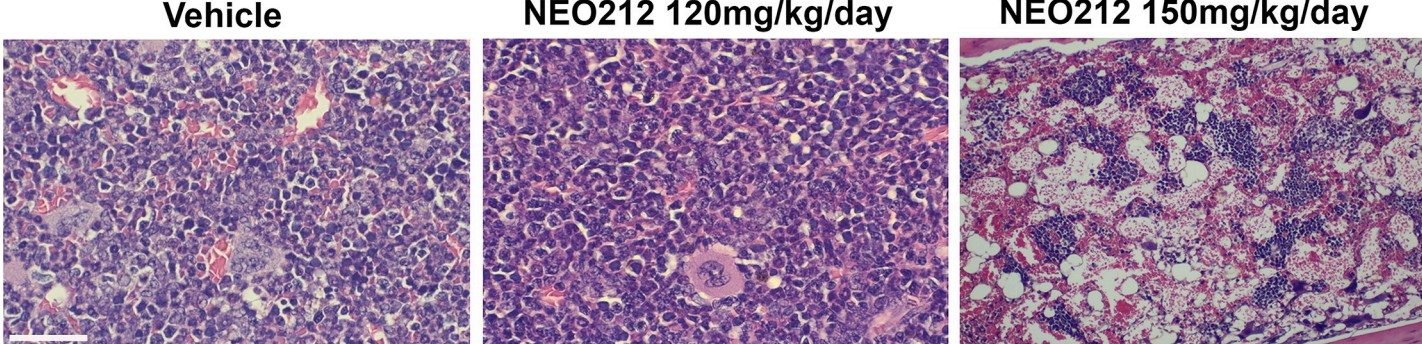

**Fig 11. Bone marrow toxicity following dose escalation studies with NEO212.** NEO212 appears to be well tolerated by the animals with no bone marrow toxicity observed at clinically relevant doses (i.e., 50 mg/kg/day) given orally over 2 weeks (using a schedule of administration of 5 days on/2 days off). Significant changes in bone marrow cellularity and morphology by H&E staining are observed in animals dosed with NEO212 at 150 mg/kg/day, but not for doses up to 120 mg/kg/day.

The higher bioavailability of NEO212 in tumoral (and potentially healthy) tissues raises the possibility that the enhanced cytotoxic and radiosensitization potencies of this drug may come at the expense of additional off-target toxicities. However, we found the drug to be well tolerated by animals dosed for up to 2 weeks with the drug as they only suffered minor weight changes (S11 Fig).

Importantly, the animals dosed with up to 120 mg/kg/day of NEO212 appeared to be spared of bone marrow toxicities over the short course of this study based on the morphological appearance and cellularity of their marrows on H&E (Fig 11). This is an important finding which indicates that the dosages at which off-target toxicities with NEO212 become apparent might be in fact much higher than the equivalent dosages translated from the Stupp protocol in mice (i.e., 25–50 mg/kg daily).

## Discussion

No direct evidence is available to demonstrate that TMZ can radiosensitize GB tumors in the clinic. However, a number of previous studies conducted with cell lines or animal models of GB suggest that TMZ possesses real albeit somewhat limited radiosensitization properties in the MGMT-deficient setting [17,18]. Historically, by chemically modifying the alkyl group that is transferred to the DNA, more potent analogs of TMZ were further developed [6], but at the expense of additional toxicities. Derivatizations of the TMZ molecule were also attempted [6] in order to improve on the tumor penetration of the drug but these also led to rather modest translational successes thus far.

In an effort to circumvent the above issues, we identified a new derivative called NEO212—i.e., a conjugation of TMZ with perillyl alcohol. In the present study, we demonstrate that NEO212 has superior tumor cell uptake and much better radiosensitization properties than TMZ at clinically relevant concentrations for GB and irrespective of MGMT or MMR status. However, we also show that the cytotoxic effects of NEO212 are entirely dependent on its alkylating properties. This statement is strongly supported by two observations: (i) once the molecule is decayed by pre-incubation in aqueous solutions, its cytotoxicity is completely lost, and (ii) as is the case with TMZ, the cytotoxic effects of NEO212 as monotherapy are hampered by the presence of MGMT or in the MMR-deficient setting. Therefore, the radiosensitization profile of NEO212 can be entirely attributed to its superior tumor uptake properties rather than a novel mechanism of cytotoxicity.

Generally, the radiosensitivity of human cells varies throughout the cell cycle while various DNA repair pathways are differentially activated during cell cycle and target different DNA lesion types with different specificities [38]. Since the DNA repair pathways rapidly remove radiation-induced DNA lesions to protect tumor and normal cells from lethality, the pharmacological manipulation of these pathways (i.e., toward radiosensitization in tumor cells and radioprotection in normal tissues) could potentially lead to further improvements in the radiation therapeutic ratio. One way to pharmacologically manipulate these repair pathways is to improve on the differential tumor uptake of alkylating agents by creating novel chemical entities with enhanced tumor bioavailability. When administered in combination with IR, once an optimal concentration of a DNA alkylating drug is achieved intracellularly, this scenario has the potential to overwhelm certain repair pathways (such as the BER system in the case of DNA methylating agents) which could in theory lead to enhanced radiosensitization effects independent of the MGMT or MMR status. Our CFA data clearly demonstrate this effect with TMZ—i.e., supraphysiologic concentrations of TMZ can radiosensitize MGMT-proficient GB cells lines.

The complexity of DNA damage can vary widely with the radiation dose (i.e., the total absorbed dose and the dose rate) and the type of chemotherapy given concurrently, ranging from isolated single-strand breaks (SSBs) or double-strand breaks (DSBs) to complex combinations of lesions arising at multiple sites [39]. SSBs generated by IR are repaired by the single strand break repair (SSBR) process which also employs the core BER enzymes (i.e., Polβ and DNA Ligase III in concert with scaffolding protein XRCC1) [37]. DSBs are of paramount importance in radiation oncology as these lesions can be induced by ionizing radiation and most chemotherapeutic agents. Three major pathways exist to repair DSBs: two are recombinogenic and require DNA resection (i.e., homologous recombination (HR) and alternative non-homologous end joining (Alt-NHEJ)) while the third one is mutagenic (i.e., non-homologous end joining (NHEJ)) [40–42]. Tumors driven by oncogenes are under constant replicative stress which introduces DNA strand breaks and other lesions at replication forks and drives the steady state level of DNA damage in these tissues to much higher levels. This renders tumor cell populations caught by genotoxic insults in the S phase of cell cycle very vulnerable to the effects of chemotherapy. Therefore, chemotherapy (such as DNA methylating agents) could represent an excellent radiosensitization solution for eliminating tumor cells in S phase. While the above provides a strong rationale for using chemotherapeutic drugs concurrently with radiotherapy, unfortunately this strategy is also thought to eventually select for and enrich tumors in slow proliferating and/or growth arrested cells with superior motility. These tumor cells are also believed to be much less reliant on recombinogenic mechanisms of DNA repair.

To unlock the radiosensitization properties of NEO212, we first measured the clonogenic survival of GB cells using the colony formation assay (CFA), which is a surrogate assay that estimates residual tumor burden after cytotoxic treatments [43]. In all these CFA analyses, we employed a treatment strategy designed to mimic the Stupp protocol. Second, using the same treatment schedules and dosages from CFA analyses, we further documented the DNA damage inflicted by NEO212 or TMZ in combination with IR at the molecular level and in a more quantitative manner by high-content widefield immunocytochemistry. This was accomplished by measuring the total amount of DNA damage by γH2AX staining [35]. The formation of γH2AX foci is one of the hallmarks of DSB signaling and is defined by the rapid phosphorylation of the histone variant H2AX at serine 139, usually executed by the ATM/ATR DNA repair kinases. This phosphorylation event sets in motion a cascade of additional repair events as it facilitates the recruitment and docking of key repair proteins such as DNA-PKcs, Rad51, Nbs1, and BRCA1 at the site of DNA damage [44,45]. The staining intensity of γH2AX foci becomes maximal at 30–40 minutes after a genotoxic insult such as IR and, as the repair

process of damaged DNA is completed, the intensity of this signal completely returns to background levels at 24 hours after radiation exposure [46,47]. On the other hand, persistent γH2AX staining at 24 hours after the genotoxic event is strongly suggestive of irreparable DNA damage [36,47] and, when quantified by high-content microscopy, it can reliably corroborate other analyses such as the clonogenic survival. Historically, the γH2AX staining has been used as a readout of DNA damage in numerous preclinical studies but also in the clinic [48,49]. In addition to our qICC studies, we used the same γH2AX staining method to make further correlations between the amount of cell death inflicted by our treatment schedules and dosages (measured by Annexin V staining) and the accumulation of persistent DSB foci (measured by γH2AX staining) in a series of intracellular FACS analyses.

Collectively, all these data suggest that the superior radiosensitization activity of NEO212 might mechanistically stem from the ability of this novel agent to generate, at drug concentrations relevant for brain tumors, levels of non-$O^6$-methylguanine lesions (i.e., N-methylpurine adducts) that are high enough to allow for synergisms with IR. Although we have not directly measured the extent of DNA methylation events generated by NEO212 vs. TMZ at clinically relevant concentrations, the generation of high levels of N-methylpurine adducts by NEO212 in this concentration range remains the only reasonable mechanistic explanation that can be inferred from our observations. The enzymatic apparatus of BER/SSBR is extremely well conserved across phyla and it evolved to become the first line of DNA repair in eukaryotes where it monitors and executes the repair of commonly occurring DNA insults (i.e., the methyl and oxidative flavors of DNA damage). While the methyl and oxidative nucleobase damage are repaired via incisional SSB (i.e., BER), the other commonly occurring SSBs (the direct SSBs), which result from linear energy transfer in the case ionizing radiation, are repaired via SSBR. Up to 90% of the non-$O^6$-methylguanine adducts generated by TMZ or NEO212 are BER substrates which theoretically should be rapidly and efficiently repaired by this system. However, when the level of N-methylpurine damage exceeds a certain threshold [8], an inherent weakness of the BER system becomes apparent as the overwhelming of the repair system itself could lead to a buildup of toxic intermediates which are normally dealt with by PARylation [50,51]. When this threshold is reached, it is believed that the accumulation of BER intermediates could lead to the collapse of replication forks and the buildup of DSBs that ensues could ultimately overwhelm the recombinogenic repair [51]. Along these lines, the data we generated with the PARP inhibitor Olaparib, which was previously shown to trap PARP-1/2 to DNA and impede the function of the BER system leading to an accumulation of toxic BER intermediates, seems to, at least partially, confirm this hypothesis. It is also true that the involvement of PARylation in DNA repair goes far beyond BER as this post-translational modification is also critical to DSB repair itself. Therefore, additional in vitro experiments with NEO212 and PARPi in which the levels of BER intermediates are rigorously monitored and quantified using sensitive assays [50] are further warranted. Nonetheless, the fact that the addition of minimally cytotoxic amounts of PARPi to clinically relevant concentrations of NEO212 has such an impressive effect along with the potential of PARPi itself for additional synergisms with radiation are all very exciting findings. Clinical trials with PARPi have generally been disappointing because of unexpected toxicities possibly due to the fact that the same high doses needed for PARP inhibition when PARPi were used as monotherapy were also implemented in combinatorial trials [37]. The fact that very low concentrations of PARPi seem to be enough to boost the radiosensitization effects of NEO212 could represent a departure from this. PARP-2 is also essential for hematopoietic stem/progenitor cell survival [52] and since most PARPi are dual PARP-1/2 inhibitors, the potential requirement for much lower doses of PARPi when combined with NEO212 could also be critical for the mitigation of important hematopoietic toxicities. Importantly, because NEO212 acts as a prodrug of TMZ, it transfers the same well-

tolerated alkyl groups by non-diseased tissues and is expected to have a similar toxicity profile as TMZ. Further in vivo studies will be however needed to confirm the radiosensitization properties of NEO212 in tumor models of TMZ-resistant GB. If validated by rigorous safety studies, the superior radiosensitization properties of NEO212 are expected to offer a much better therapeutic solution for newly diagnosed GB patients, irrespective of the MGMT or MMR status of their tumors. This could represent a significant improvement over the Stupp protocol. However, we predict that patients diagnosed with GB tumors with methylated MGMT promoters will still probably benefit the most from a NEO212-based chemoradiation regimen.

Lastly, bone marrow and brain toxicity studies—including the measuring of the amount of DNA damage in non-diseased brain areas along with markers of neuroinflammation—in non-tumor and tumor bearing animals will be required to further validate the toxicity profile of NEO212. Although our preliminary in vivo toxicity studies conducted with NEO212 seem to argue against this point, due to its higher cellular uptake compared to TMZ, NEO212 might prove to have a narrower therapeutic index than TMZ in vivo, particularly in the bone marrow compartment in humans. Therefore, a more direct quantification of the differences in alkylation capacity between NEO212 and TMZ at equivalent but clinically relevant dosages will be warranted in order to further validate our current observations with this molecule. One sensitive method that allows to directly measure the levels of DNA alkylation by NEO212 and TMZ requires the labeling of both molecules with a $^{14}$C radioisotope (i.e., specifically at the methyl group that is transferred to DNA via the methyldiazonium ion). This approach is expected to generate a far more precise readout of the true extent of DNA methyl damage inflicted by NEO212 to various organs. Along these lines, we are currently planning to conduct an in vivo tissue methylation study in which alkylation of target organs harvested from non-tumor and tumor bearing animals dosed with either $^{14}$C-NEO212 or $^{14}$C-TMZ will be quantified in a liquid scintillation counter.

In summary, conjugation of perillyl alcohol to TMZ, resulting in the novel agent NEO212, generated a molecule with intriguing promise for clinical applications, in particular in view of its pronounced radiosensitization potential. In current clinical practice, radiosensitization by TMZ is significantly blunted, because achievable drug concentrations within brain tumors do not exceed 10 μM. In comparison, our study established that, at similarly low concentrations, NEO212 is able to unfold its radiosensitization activity to achieve substantially greater tumor cell killing. Moreover, NEO212 appears to enable greater cytotoxic impact of PARPi, possibly paving the way for these inhibitors to finally display their long-awaited therapeutic impact in the clinical setting. Within the chemoradiation schedule of the Stupp protocol, we predict that replacement of TMZ with NEO212 might achieve better therapeutic outcomes not only for patients with methylated MGMT promoter, but also for those with unmethylated MGMT promoter, where the addition of TMZ has shown only marginal benefit over radiation alone. It will be important to further develop NEO212 and validate these assertions in the clinic.

## Supporting information

**S1 Fig. The maps of the DNA constructs used to generate the TMZ-resistant GB cell variants.** The LN229M and U251M cell variants were generated after infection with a lentiviral construct that carries the CDS for human MGMT in tandem with the one for firefly luciferase separated by an IRES sequence and under an EF1α promoter and, in a separate ORF on the same lentivector, the CDS for enhanced GFP under a CMV promoter (Panel A). The GFP reporter facilitated the selection of high MGMT expressing variants of these GB cells which were sorted to purity by FACS. The GL261M cell variant was generated after transfection with a plasmid construct that carries the murine Mgmt CDS under the control of a murine

phosphoglycerate kinase (mPGK) promoter (Panel B). The transfected cells were further enriched in murine Mgmt by serial passaging in medium containing increasing concentrations of TMZ.
(TIF)

**S2 Fig. NEO212 can radiosensitize TMZ-resistant GB cells at relevant concentrations.** In these CFA analyses, we tested both drugs using a treatment schedule designed to mimic the Stupp protocol (i.e., five consecutive days of concurrent chemotherapy plus radiotherapy vs. five consecutive days of monotherapies). The colony survival data show that NEO212 can synergize with ionizing radiation in the clinically relevant concentration range (i.e., 10 μM or less), whereas TMZ becomes synergistic only outside (i.e., >10 μM) of this concentration range where presumably achieves levels of DNA alkylation optimal for synergistic effects to take place (* indicates a p<0.01 determined by Student's t-test).
(TIF)

**S3 Fig. The cytotoxicity and radiosensitization properties of NEO212 are fully dependent on its DNA alkylating activity.** If NEO212 is pre-incubated in medium for 24hrs (i.e., by becoming decayed NEO212 or dNEO212) before is added to the cells, it loses all of its cytotoxic and radiosensitization properties. In these CFA analyses, we tested dNEO212 using the same treatment schedule designed to mimic the Stupp protocol (i.e., five consecutive days of treatments). The colony survival data show that dNEO212 completely lost its cytotoxicity even against TMZ-sensitive cells (* indicates a p<0.01 determined by Student's t-test).
(TIF)

**S4 Fig. The cytotoxicity and radiosensitization properties of NEO212 are fully dependent on its DNA alkylating activity.** If NEO212 is pre-incubated in medium for 24hrs (i.e., by becoming decayed NEO212 or dNEO212) before is added to the cells, it loses all of its cytotoxic and radiosensitization properties. In these CFA analyses, we tested dNEO212 using the same treatment schedule designed to mimic the Stupp protocol (i.e., five consecutive days of treatments). The colony survival data show that dNEO212 completely lost its cytotoxicity even against TMZ-sensitive cells (* indicates a p<0.01 determined by Student's t-test).
(TIF)

**S5 Fig. Quantitative ICC analysis of irreparable DNA damage in LN229TR2/T98G GB cells.** LN229TR2 (a MMR-deficient variant of LN229 cells) and T98G (an endogenously expressing MGMT line) cells were seeded at high densities (50,000 cells/cm$^2$) and either left untreated (UT) or treated for five consecutive days with either 10 μM TMZ (T) or NEO212 (N) or 2 Gy (R) alone or combinations without (T+R or N+R) or with (T+R+O or N+R+O) Olaparib (O). The cells were probed with a γH2AX antibody and an AF647-labeled secondary and nuclei were counterstained with DAPI. Persistent γH2AX foci (red) were digitally counted relative to the total number of cell nuclei (blue). Each panel is data dense and represents a composite of 36 fields in total (i.e., a square of about 3x3 mm) captured on a widefield microscopy instrument and digitally stitched together. Scale bar is 500 μm (upper left corner).
(TIF)

**S6 Fig. Quantitative ICC analysis of irreparable DNA damage in GL261/GL261M GB cells.** Cells were seeded at high densities (50,000 cells/cm$^2$) and either left untreated (UT) or treated for five consecutive days with either 10 μM TMZ (T) or NEO212 (N) or 2 Gy (R) alone or combinations without (T+R or N+R) or with (T+R+O or N+R+O) Olaparib (O). The cells were probed with a γH2AX antibody and an AF647-labeled secondary and nuclei were counterstained with DAPI. Persistent γH2AX foci (red) were digitally counted relative to the total

number of cell nuclei (blue). Each panel is data dense and represents a composite of 36 fields in total (i.e., a square of about 3x3 mm) captured on a widefield microscopy instrument and digitally stitched together. Scale bar is 500 μm.
(TIF)

**S7 Fig. The PARP inhibitor Olaparib is minimally cytotoxic at low nanomolar concentrations.** In these CFA analyses, we tested Olaparib on all GB cell lines over a range of concentrations (0–1000 nM). The colony survival data show that Olaparib is barely cytotoxic in the lower end of this concentration range.
(TIF)

**S8 Fig. Olaparib does not synergize with IR at minimally cytotoxic concentrations.** Cells were seeded at high densities (50,000 cells/cm$^2$) and either left untreated or treated for five consecutive days with either 10 nM of Olaparib (O) or 2 Gy (R) alone or the combination of both (R+O). Persistent γH2AX foci were digitally counted and expressed relative to the total number of cell nuclei. All ratio values (pixel counts from treatments expressed as a % ratio of persistent γH2AX foci to nuclei values normalized to untreated counts) were found statistically significant with p values of <0.01 (ANOVA with Tukey post-hoc testing). A fold sensitization value of 1 or close to 1 signifies additive, non-synergistic effects. We deliberately kept the scale of the y-axis the same size for all γH2AX quantifications throughout the study for comparison purposes.
(TIF)

**S9 Fig. FACS analysis of cell death resulting from irreparable DNA damage.** LN229TR2 and T98G cells were seeded at high densities (50,000 cells/cm$^2$) and were either left untreated (UT) or treated for five consecutive days with either 10 μM TMZ (T) or NEO212 (N) or 2 Gy (R) alone or combinations without (T+R or N+R) or with (T+R+O or N+R+O) Olaparib (O). The cells were probed with a Pacific Blue-labeled Annexin V and then fixed, permeabilized and probed with an AF488-labeled γH2AX antibody. γH2AX/Annexin V double positive cells (i.e., dead cells due to irreparable DNA damage) are shown as percentages of total cell numbers. Representative panels are shown from three independent experiments.
(TIF)

**S10 Fig. FACS analysis of cell death resulting from irreparable DNA damage.** GL261 and GL261M cells were seeded at high densities (50,000 cells/cm$^2$) and were either left untreated (UT) or treated for five consecutive days with either 10 μM TMZ (T) or NEO212 (N) or 2 Gy (R) alone or combinations without (T+R or N+R) or with (T+R+O or N+R+O) Olaparib (O). The cells were probed with a Pacific Blue-labeled Annexin V and then fixed, permeabilized and probed with an AF488-labeled γH2AX antibody. γH2AX/Annexin V double positive cells (i.e., dead cells due to irreparable DNA damage) are shown as percentages of total cell numbers. Representative panels are shown from three independent experiments.]
(TIF)

**S11 Fig. Mouse weight curves during dose escalation studies with NEO212.** NEO212 appears to be well tolerated by the animals with minor weight loss observed at clinically relevant doses (i.e., 50 mg/kg/day) given orally over 2 weeks (using a schedule of administration of 5 days on/2 days off). At higher dosages (i.e., 150 mg/kg/day) of NEO212, the change in weight appears to be more pronounced, with a loss of about 10% of the total body weight at the beginning of the study.
(TIF)

**S1 File.**
(PDF)

## Acknowledgments

We thank Thu Zan Thein from the Department of Neurological Surgery at USC for expert help with stereotactic tumor implantations and animal studies, Bernadette Masinsin from the Flow Cytometry Core at USC for technical assistance and expert help with FACS analyses, and Alan Epstein (USC) for the mouse glioma cell line.

## Author Contributions

**Conceptualization:** Radu O. Minea, Thomas C. Chen.

**Data curation:** Radu O. Minea, Tuan Cao Duc, Stephen D. Swenson, Hee-Yeon Cho, Mickey Huang, Hannah Hartman, Florence M. Hofman.

**Formal analysis:** Radu O. Minea, Tuan Cao Duc, Stephen D. Swenson, Hee-Yeon Cho, Mickey Huang, Florence M. Hofman, Axel H. Schönthal.

**Funding acquisition:** Thomas C. Chen.

**Methodology:** Radu O. Minea, Stephen D. Swenson, Hee-Yeon Cho.

**Project administration:** Radu O. Minea.

**Supervision:** Radu O. Minea.

**Validation:** Radu O. Minea, Stephen D. Swenson.

**Visualization:** Radu O. Minea.

**Writing – original draft:** Radu O. Minea.

**Writing – review & editing:** Radu O. Minea, Axel H. Schönthal, Thomas C. Chen.

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
