## [Decision Letter · Decision Letter 0]

4 Jun 2020

PONE-D-20-13771

Developing a clinically relevant radiosensitizer for temozolomide-resistant gliomas

PLOS ONE

Dear Dr. Minea,

Thank you for submitting your manuscript to PLOS ONE. After careful consideration, we feel that it has merit but does not fully meet PLOS ONE’s publication criteria as it currently stands. Therefore, we invite you to submit a revised version of the manuscript that addresses the points raised during the review process.

As detailed below, both reviewers have suggested numerous modifications that would be critical to completion of this manuscript. I would ask that you address all the concerns raised experimentally and textually. 

We look forward to receiving your revised manuscript.

Kind regards,

Robert W Sobol, PhD

Academic Editor

PLOS ONE

Journal Requirements:

'I have read the journal's policy and the authors of this manuscript have the following

competing interests: T.C.C. is the founder of NeOnc Technologies, Inc., a startup

company formed to develop derivative drug-conjugate solutions for intranasal, oral,

and systemic administration in multiple clinical indications. R.O.M., S.D.S., and T.C.C.

are also co-founders of Disintegrin Therapeutics, Inc., a startup company aimed to

develop disintegrin-based theranostics for multiple clinical indications.'

Additional Editor Comments (if provided):

Reviewers' comments:

Reviewer's Responses to Questions

**Comments to the Author**

1. Is the manuscript technically sound, and do the data support the conclusions?

Reviewer #1: Partly

Reviewer #2: Yes

2. Has the statistical analysis been performed appropriately and rigorously? 

Reviewer #1: Yes

Reviewer #2: N/A

3. Have the authors made all data underlying the findings in their manuscript fully available?

Reviewer #1: Yes

Reviewer #2: Yes

4. Is the manuscript presented in an intelligible fashion and written in standard English?

Reviewer #1: Yes

Reviewer #2: Yes

5. Review Comments to the Author

Reviewer #1: The manuscript presents data on a structural analog of temozolomide called NEO212 that is a conjugate of TMZ and perillyl alcohol. The study is highly relevant because although TMZ and radiation is standard of care for glioblastoma, the long-term prognosis remains grim. The authors present data that NEO212 is more active in vitro cell culture models than TMZ alone. The authors present data about the uptake of TMZ and NEO212, and show that NEO212 accumulates in cells, brain and tumor tissue to higher levels than TMZ. Although there is interesting and promising data shown, there is a concern with the major premise that could be addressed more rigorously. The premise that NEO212 has enhanced activity because it is producing more DNA adducts, specifically N-methypurine adducts, which then overwhelms BER and can also radiosensitize. It is a reasonable assumption but it is inferential because methyl purine adducts have not been measured in this or other previous studies. Related, a key sentence stating the adduct distribution for TMZ and NEO212 is the same, in lines 313-315 goes unreferenced. Moreover, the reference to synthesis and characterization of NEO212 elsewhere in the manuscript, #28, is actually a review with some in silico analysis but no apparent in vitro or cellular analysis of DNA base adduct distribution. The authors should provide the data or clearly cite a reference, because this is at the core of argument for the mechanism of action for NEO212, its enhanced penetration into cells and across the BBB. This data would enhance the rigor supporting the hypothesis. The experiments with “decayed” NEO212 in supplementary Figures S3 and S4 are insufficiently explained.

The data presented in the manuscript supporting DNA damage, immunofluorescence of gamma-H2AX after a 24 hour time point is inferential. The images in Figures 5 and 6 are poor quality, may just be the PDF. The pixilation does not provide confidence in the quantitation shown in Figure 7. The Figure legend, lines 451 and 459, state scale bar is 200 microns, but there are no scale bars on the images.

Strengthening the rigor of the supporting data would give more confidence in the potential impact of exploring NEO212 as a novel treatment for glioblastoma.

Other comments.

There is a line in the figure legend of Figure 3 that does not make sense, nor does the corresponding data in Figure 3 with O6BG. Line 374-76. "Inclusion of O6BG improved the survival." Benzylguanine covalently inactivates the MGMT protein. This inactivation should SENSITIZE cells because there are hundreds of papers that show that MGMT protein activity is protective against the methyl damage caused by TMZ and mechanistically similar methylating agents. Or O6BG has no effect in cells lacking MGMT.

Line 333, Figure 2 does not have a figure legend.

Minor comments.

Line 98, MGMT is a suicide protein, not an enzyme.

Line 105. “potentially” or “potently”? The cytotoxicity of O6-methylguanine is well established.

Reviewer #2: The authors studied the antitumor activity of a novel chemical entity, NEO212—a derivatization of TMZ generated by coupling TMZ to perillyl alcohol, a natural monoterpene. Compared to temozolomide (TMZ), tumor cell uptake of NEO212 more than that of TMZ. In mouse models, NEO212 was more efficient than TMZ at crossing the blood-brain barrier, and it preferentially accumulated in tumor tissue over normal brain tissue. The in vitro analyses revealed NEO212 showed significantly more potent cytotoxic and radiosensitizing properties than TMZ at clinically relevant concentrations. Overall, the manuscript is well written, and the data support the conclusion well. Because the study involved in mouse model, a tumor growth curve, mouse survival curve or endpoint tumor size images comparing TMZ or NEO211 treatment is needed to support the conclusion better. The mouse body weight curve is needed to indicate the systemic toxicity.

6. PLOS authors have the option to publish the peer review history of their article (what does this mean?). If published, this will include your full peer review and any attached files.

Reviewer #1: No

Reviewer #2: No

---

## [Author Response · Author response to Decision Letter 0]

20 Jul 2020

We would like to refer the Academic Editor and the Reviewers to read the included Response to Reviewers file. Thank you.

---

## [Decision Letter · Decision Letter 1]

13 Aug 2020

Developing a clinically relevant radiosensitizer for temozolomide-resistant gliomas

PONE-D-20-13771R1

Dear Dr. Minea,

We’re pleased to inform you that your manuscript has been judged scientifically suitable for publication and will be formally accepted for publication once it meets all outstanding technical requirements.

Kind regards,

Robert W Sobol, PhD

Academic Editor

PLOS ONE

Additional Editor Comments (optional):

Reviewers' comments:

Reviewer's Responses to Questions

**Comments to the Author**

1. If the authors have adequately addressed your comments raised in a previous round of review and you feel that this manuscript is now acceptable for publication, you may indicate that here to bypass the “Comments to the Author” section, enter your conflict of interest statement in the “Confidential to Editor” section, and submit your "Accept" recommendation.

Reviewer #1: All comments have been addressed

Reviewer #2: All comments have been addressed

2. Is the manuscript technically sound, and do the data support the conclusions?

Reviewer #1: (No Response)

Reviewer #2: Yes

3. Has the statistical analysis been performed appropriately and rigorously? 

Reviewer #1: (No Response)

Reviewer #2: Yes

4. Have the authors made all data underlying the findings in their manuscript fully available?

Reviewer #1: (No Response)

Reviewer #2: Yes

5. Is the manuscript presented in an intelligible fashion and written in standard English?

Reviewer #1: (No Response)

Reviewer #2: Yes

6. Review Comments to the Author

Reviewer #1: The authors have addressed the major critiques. They have included preliminary in vivo tox studies that seem promising. It is noted that there still isn't direct evidence that NEO212 has a similar adduct distribution to TMZ. However, the totality of the evidence provided in the manuscript makes this a minor point, and they propose this as future work.

Reviewer #2: My concerns have been addressed. The safety of the drug has been added. Due to the Covid-19, the other missing data of the mouse experiment might be forgivable.

7. PLOS authors have the option to publish the peer review history of their article (what does this mean?). If published, this will include your full peer review and any attached files.

Reviewer #1: No

Reviewer #2: No

---

## [Editor Report · Acceptance letter]

19 Aug 2020

PONE-D-20-13771R1 

Developing a clinically relevant radiosensitizer for temozolomide-resistant gliomas 

Dear Dr. Minea:

I'm pleased to inform you that your manuscript has been deemed suitable for publication in PLOS ONE. Congratulations! Your manuscript is now with our production department. 

Kind regards, 

on behalf of

Dr. Robert W Sobol 

Academic Editor

PLOS ONE